# Beyond Topological Self-Explainable GNNs:
# A Formal Explainability Perspective

**Steve Azzolin** [* 1]  **Sagar Malhotra** [* 2]  **Andrea Passerini** [1]  **Stefano Teso** [1]

## Abstract

Self-Explainable Graph Neural Networks (SE-GNNs) are popular explainable-by-design GNNs, but their explanations' properties and limitations are not well understood. Our first contribution fills this gap by formalizing the explanations extracted by some popular SE-GNNs, referred to as Minimal Explanations (MEs), and comparing them to established notions of explanations, namely Prime Implicant (PI) and faithful explanations. Our analysis reveals that MEs match PI explanations for a restricted but significant family of tasks. In general, however, they can be less informative than PI explanations and are surprisingly misaligned with widely accepted notions of faithfulness. Although faithful and PI explanations are informative, they are intractable to find and we show that they can be prohibitively large. Given these observations, a natural choice is to augment SE-GNNs with alternative modalities of explanations taking care of SE-GNNs' limitations. To this end, we propose Dual-Channel GNNs that integrate a white-box rule extractor and a standard SE-GNN, adaptively combining both channels. Our experiments show that even a simple instantiation of Dual-Channel GNNs can recover succinct rules and perform on par or better than widely used SE-GNNs.

## 1. Introduction

***Self-Explainable GNNs*** (SE-GNNs) are Graph Neural Networks (Scarselli et al., 2008; Wu et al., 2020b) designed to combine high performance and *ante-hoc* interpretability. In a nutshell, a SE-GNN integrates two GNN modules: an ***explanation extractor*** responsible for identifying a class-discriminative subgraph of the input and a ***classifier*** mapping said subgraph onto a prediction. Since this subgraph, taken in isolation, is enough to infer the prediction, it plays the role of a local explanation thereof. Despite the popularity of SE-GNNs (Miao et al., 2022a; Lin et al., 2020; Serra & Niepert, 2022; Zhang et al., 2022; Ragno et al., 2022; Dai & Wang, 2022), little is known about the properties and limitations of their explanations. Our work fills this gap.

Focusing on graph classification, we introduce the notion of Minimal Explanations (MEs) as the minimal subgraphs of the input ensuring that the classifier outputs the target prediction. We then show that some popular SE-GNNs are implicitly optimized for generating MEs. We further compare MEs with two other families of formal explanations: ***Prime Implicant*** explanations[1] (PIs) and ***faithful*** explanations. Faithful explanations are subgraphs that are *sufficient* and *necessary* for justifying a prediction, i.e., they capture all and only those elements that cause the predicted label (Yuan et al., 2022; Tan et al., 2022; Agarwal et al., 2023; Azzolin et al., 2025). PIs, instead, are minimally sufficient explanations extensively studied in formal explainability of tabular and image data (Marques-Silva, 2023; Darwiche & Hirth, 2023; Wang et al., 2021). They are also highly informative, e.g., for any propositional formula the set of PIs is enough to reconstruct the original formula (Ignatiev et al., 2015). Moreover, both PI and sufficient explanations are tightly linked to counterfactuals (Beckers, 2022) and adversarial robustness (Ignatiev et al., 2019). An example of these families' nuances is shown in Fig. 1.

Our results show that MEs match PI explanations in the restricted but important family of ***motif-based prediction tasks***, where labels depend on the presence of topological motifs. Although in these tasks MEs inherit all benefits of PIs, in general they are neither faithful nor PIs.

These observations motivate augmenting SE-GNNs with alternative explanation modalities to address their limitations. To this end, we introduce ***Dual-Channel GNNs*** (DC-GNNs), a novel family of SE-GNNs that aim to extend the perks of motif-based tasks to more general settings. DC-GNNs combine a SE-GNN and a non-relational white-box

---

[*]Equal contribution  [1]DISI, University of Trento, Trento, Italy  [2]TU Wien, Wien, Austria. Correspondence to: Steve Azzolin <steve.azzolin@unitn.it>, Sagar Malhotra <sagar.malhotra@tuwien.ac.at>.

*Proceedings of the $42^{nd}$ International Conference on Machine Learning*, Vancouver, Canada. PMLR 267, 2025. Copyright 2025 by the author(s).

---

[1]Also known as sufficient reasons (Darwiche & Hirth, 2023).

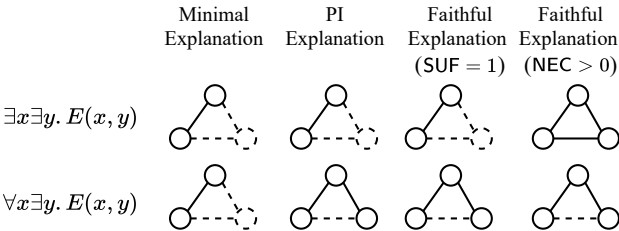

|  | Minimal Explanation | PI Explanation | Faithful Explanation (SUF = 1) | Faithful Explanation (NEC > 0) |

Figure 1: Examples of a Minimal, PI, and faithful explanation (w.r.t Proposition 5.3 and Proposition 5.4) for the predictions of the two classifiers $\exists x \exists y.E(x,y)$ and $\forall x \exists y.E(x,y)$ introduced in Section 2 on a triangle. Solid nodes and edges represent the explanation $R$, whilst dashed ones represent the complement $C = G \setminus R$. At a high level, MEs highlight the smallest label-preserving subgraphs, PIs and faithful sufficient explanations encode the subgraph ensuring that no complement perturbation can change the prediction, while faithful necessary explanations highlight subgraphs that once removed bring a prediction change.

predictor, adaptively employing one or both depending on the task. Intuitively, the non-relational channel handles non-topological aspects of the input, leaving the SE-GNN free to focus on topological motifs with MEs. This setup encourages the corresponding MEs to be more compact, all while avoiding the (generally exponential (Marques-Silva, 2023)) computational cost of extracting PIs explicitly. Empirical results on three synthetic and five real-world graph classification datasets highlight that DC-GNNs perform as well or better than SE-GNNs by adaptively employing one channel or both depending on the task.

**Contributions**. Summarizing, we:

- Formally characterize the class of explanations that a popular family of SE-GNNs optimizes for, namely MEs.

- Show that MEs share key properties of PI and faithful explanations for motif-based prediction tasks.

- Propose Dual-Channel GNNs and compare them empirically to representative SE-GNNs, highlighting their promise in terms of explanation size and performance.

## 2. Preliminaries

We will use $g$ to represent a **graph classifier**, usually representing a GNN. Given a graph $G = (V, E)$, we will use $g(G)$ to denote $g$'s predicted label on $G$. Additionally, graphs can be annotated with edge or node features. Throughout, we will use the notation $R \subset G$ to denote that $R$ is a **subgraph** of $G$ and $R \subseteq G$ to denote that $R$ may be a subgraph or $G$ itself. Note that we assume that a subgraph

$R \subseteq G$ can contain all, less, or none of the features of the nodes (or edges) in $G$. We will use $|G|$ to indicate the size of the graph $G$. The size may be defined in terms of the number of nodes, edges, features, or their combination, and the precise semantics will be clear from the context.

**Self-explainable GNNs**. SE-GNNs are designed to complement the high performance of regular GNNs with *ante-hoc* interpretability. They pair an ***explanation extractor*** $q$ mapping the input $G$ to a subgraph $q(G) = R \subseteq G$ playing the role of a local explanation, and a ***classifier*** $f$ using $R$ to infer a prediction:

$$g(G) = f(q(G)) \tag{1}$$

In practice, the explanation extractor $q$ outputs per-edge ***relevance scores*** $p_{uv} \in \mathbb{R}$, which are translated into an edge-induced subgraph $R$ via thresholding (Yu et al., 2022) or top-$k$ selection (Miao et al., 2022a; Chen et al., 2024). While we focus on per-edge relevance scores due to their widespread use, our results equally apply to per-node relevance scores.

Approaches for training SE-GNNs are designed to extract an interpretable subgraph that suffices to produce the target prediction, often formalized in terms of sparsity regularization (Sparsity) (Lin et al., 2020; Serra & Niepert, 2022) or the Information Bottleneck (IB) (Tishby et al., 2000). Since the exact IB is intractable and difficult to estimate (Kraskov et al., 2004; McAllester & Stratos, 2020), common approaches devise bounds on the divergence between the relevance scores and an uninformative prior controlled by the parameter $r \in [0, 1]$ (Miao et al., 2022a;b). This work focuses on representative training objectives for both Sparsity- and IB-based SE-GNNs, indicated in Table 1.

**Logical classifiers**. To prove our theoretical results, we will use basic concepts from First-Order Logic (FOL) as described in Barceló et al. (2020) and Grohe (2021), and use $E(x, y)$ to denote an undirected edge between $x$ and $y$. A FOL boolean classifier is a FOL sentence $\Phi$ that labels an instance $G$ positively if $G \models \Phi$ and negatively otherwise. For ease of discussion, we fix two FOL classifiers that will be used to provide examples and intuition in the remainder of the paper: $\exists x \exists y.E(x, y)$ classifies a graph positively if it has an edge, while $\forall x \exists y.E(x, y)$ when it has no isolated nodes. Note that both can be expressed by a GNN, as they are in the logical fragment $C^2$ (see Theorem IX.3 in Grohe (2021)). We say that two classifiers are distinct if there exists at least one instance where their predictions differ. Although most of our results discuss general graph classifiers, they equally apply to the specific case of GNNs.

## 3. What are SE-GNN Explanations?

Despite being tailored for explainability, SE-GNNs lack a precise description of the properties of the explanations they

| MODEL | GROUP | LEARNING OBJECTIVE |
|---|---|---|
| GISST (Lin et al., 2020), SMGNN (ours) | Sparsity | $\mathcal{L} + \frac{\lambda_1}{|E|}\sum_{(u,v)\in E} p_{uv} + \frac{\lambda_2}{|E|}\sum_{(u,v)\in E} p_{uv}\log(p_{uv}) + (1-p_{uv})\log(1-p_{uv})$ |
| GSAT, LRI (Miao et al., 2022a;b), GMT (Chen et al., 2024) | IB | $\mathcal{L} + \lambda_1 \sum_{(u,v)\in E} p_{uv}\log(\frac{p_{uv}}{r}) + (1-p_{uv})\log(\frac{1-p_{uv}}{1-r})$ |

Table 1: **Training objectives of popular SE-GNNs**. $\mathcal{L}$ is the cross-entropy loss between model predictions $f(q(G))$ and the target variable $Y$. For GISST and SMGNN, the second term in the objective pushes the relevance score $p_{uv}$ of each edge to be sparse, while the last term pushes scores to align to either 0 or 1 via an entropy loss (Lin et al., 2020). For GSAT, LRI, and GMT instead, the second term is the IB regularization pushing relevance scores close to the uninformative value controlled by the hyper-parameter $r$ (Miao et al., 2022a). $\lambda_1$ and $\lambda_2$ are the relative strengths of each regularization term. Theorem 3.2 shows that all of them optimize for generating Minimal Explanations (Definition 3.1).

extract. In this section, we present a formal characterization (in Definition 3.1) of explanations extracted by SE-GNNs, called ***Minimal Explanations*** (MEs). In Theorem 3.2 we show that, for an SE-GNN with perfect predictive accuracy and a *hard* explanation extractor, the loss functions in Table 1 are minimal iff the explanation $R$ is a ME.

**Definition 3.1** (Minimal Explanations). Let $g$ be a graph classifier and $G$ be an instance with predicted label $g(G)$, then $R$ is a Minimal Explanation for $g(G)$ if:

1. $R \subseteq G$

2. $g(G) = g(R)$

3. There does not exist an explanation $R' \subseteq G$ such that $|R'| < |R|$ and $g(G) = g(R')$.

Minimal Explanations are not unique, and we use $\text{ME}(g(G))$ to denote the set of all MEs for $g(G)$.

Since our goal is to formally analyze explanations extracted by SE-GNNs, we make idealized assumptions about the classifier $f$. Hence, we assume that the SE-GNN is expressive enough and attains perfect predictive accuracy, i.e., it always returns the correct label for $G$. We also assume that it learns a *hard* explanation extractor, i.e., $q$ outputs scores saturated[2] to $\{0, 1\}$ (Yu et al., 2020). Under these assumptions, we show that SE-GNNs in Table 1 optimize for generating MEs.

**Theorem 3.2.** *Let $g$ be an SE-GNN with a ground truth classifier $f$ (i.e., $f(G)$ always returns the true label for $G$), a hard explanation extractor $q$, and perfect predictive accuracy. Then, $g$ achieves minimal true risk (as indicated in Table 1) if and only if for any instance $G$, $q(G)$ provides Minimal Explanations for the predicted label $g(G)$.*

*Proof Sketch.* For a hard explanation extractor $q$, the risk terms in Table 1 reduce to $\mathcal{L}(f(q(G)), Y) + \lambda_1 |q(G)|/|E|$ and $\mathcal{L}(f(q(G), Y) + \lambda_1 |q(G)| \log(r^{-1})$. Since $g$ attains perfect predictive accuracy, $\mathcal{L}(f(q(G)), Y)$ is minimal. Hence,

for both cases, the risk is minimized when $q(G)$ is the smallest subgraph that preserves the label, i.e., an ME. $\square$

Proofs relevant to the discussion are reported in the main text, while the others are available in Appendix A. Having established a link between SE-GNNs and MEs, we proceed to analyze the formal properties of MEs, starting from the following remark encoding a broad notion of *informative explanation*.

*Remark* 3.3. A simple desideratum for any type of explanation is that it ***gives information about the classifier beyond the predicted label***. A weak formulation of this desideratum is that explanations for two distinct classifiers should differ on at least one instance where they predict the same label[3]

Note that Remark 3.3 can be seen as the dual of the Implementation Invariance Axiom (Sundararajan et al., 2017). The following theorem, however, shows that MEs can fail to satisfy this desideratum for certain prediction tasks, indicating potential limits in the informativeness of MEs.

**Theorem 3.4.** *There exist two distinct classifiers $g$ and $g'$ such that for any $G$ where $g(G) = g'(G)$ we have that*

$$\text{ME}(g(G)) = \text{ME}(g'(G)) \tag{2}$$

*Proof.* Let $g$ and $g'$ be boolean graph classifiers given by FOL formulas $g = \exists x \exists y. E(x, y)$ and $g' = \forall x \exists y. E(x, y)$ respectively. Let $G$ be any positive instance for both $g$ and $g'$, and let $E$ and $V$ be the set of edges and nodes in $G$, respectively. For any $e \in E$, we have that $e$ is a Minimal Explanation for both $g(G)$ and $g'(G)$ (e.g. see Fig. 1). Hence, $\text{ME}(g(G)) = \text{ME}(g'(G)) = E$. Similarly, if $G$ is a negative instance for both $g$ and $g'$, we have that a Minimal Explanation is a subgraph consisting of any single node from $G$, and $\text{ME}(g(G)) = \text{ME}(g'(G)) = V$. $\square$

Intuitively, this result indicates that there exist two distinct graph classifiers for which MEs are the same for any input.

---

[2]For IB-based losses in Table 1 scores saturate to $\{r, 1\}$.

[3]To avoid this desideratum being vacuously satisfied, we assume that there exists at least one instance where the two classifiers predict the same label.

Hence, inspecting explanations alone makes it impossible to tell the two apart, meaning that MEs can fail to be informative wrt Remark 3.3.

Theorem 3.4 highlights additional insights worth discussing. Consider again the classifier $g' = \forall x \exists y.E(x,y)$, and let $G$ be a negative instance for $g'$ composed of three nodes $\{u_1, u_2, u_3\}$, only two of which are connected by the edge $(u_1, u_2) \in E$. In this case, a user may expect to see the isolated node $u_3$ as an explanation. However, any subgraph consisting of a single isolated node from the set $\{u_1, u_2, u_3\}$ is a valid ME. This highlights that as MEs focus only on the subgraphs allowing the model to reproduce the same prediction, they may lead to counter-intuitive explanations.

Furthermore, Theorem 3.4 implies that

$$\bigcup_{G \in \Omega^{(y)}} \mathrm{ME}(g(G)) = \bigcup_{G \in \Omega^{(y)}} \mathrm{ME}(g'(G)) \quad (3)$$

where $\Omega^{(y)}$ is the set of all instances $G$ such that $g(G) = g'(G) = y$, with $g = \exists x \exists y.E(x,y)$ and $g'$ is as above. Intuitively, Eq. (3) shows that the insight of Theorem 3.4 applies even when aggregating MEs across all instances where the two classifiers yield the same prediction. Hence, there are classifiers where model-level explanations built by aggregating over local explanations (Setzu et al., 2021; Azzolin et al., 2022) may also not be informative w.r.t. Remark 3.3.

In the next section, we investigate a widely accepted formal notion of explanations, namely Prime Implicant explanations (PIs). We analyze the informativeness of MEs compared to PIs and characterize when they match.

## 4. Minimal and Prime Implicant Explanations

Having established the link between SE-GNNs and MEs in Section 3, in this section, we provide a formal comparative analysis between MEs and PIs for graph classifiers. While PIs are extensively studied for tabular and image-like data (Marques-Silva et al., 2020; Marques-Silva, 2023), little investigation has been carried out for graphs. Our analysis shows that MEs match PIs for a large class of tasks – those based on the recognition of *motifs* – but they do not align in general. We also show that PIs can be more informative than MEs w.r.t the desideratum in Remark 3.3. Let us start by defining PIs for graph classifiers.

**Definition 4.1** (PI explanation). Let $g$ be a classifier and $G$ be an instance with predicted label $g(G)$, then $R$ is a Prime Implicant explanation for $g(G)$ if:

1. $R \subseteq G$.

2. For all $R'$, such that $R \subseteq R' \subseteq G$, we have that $g(G) = g(R')$.

3. No other $R' \subset R$ satisfies both (1) and (2).

Like MEs, PIs are not unique, and we use $\mathrm{PI}(g(G))$ to denote the set of all PIs for $g(G)$.

PIs feature several nice properties, in that they are guaranteed to be the minimal explanations that are provably sufficient for the prediction (Shih et al., 2018; Beckers, 2022; Darwiche & Hirth, 2023). To illustrate the difference between MEs and PIs, we provide an example for two different classifiers in Fig. 1. Note that for the classifier $\exists x \exists y.E(x,y)$, MEs match PIs. This observation indeed generalizes to all existentially quantified positive FOL formulas, as shown next.

### 4.1. MEs Match PI for Positive Existential Classifiers

We now show that our previous observation that MEs equal PIs for $\exists x \exists y.E(x,y)$ generalizes to all positive existential tasks. This reinforces the use of SE-GNNs in various practical applications and our proposed method, as discussed at the end of this section and in Section 6.

**Theorem 4.2.** *Given a classifier $g$ expressible as a purely existentially quantified positive first-order logic formula and a positive instance $G$ of any size, then a Minimal Explanation for $g(G)$ is also a Prime Implicant explanation for $g(G)$.*

*Proof sketch.* A purely existentially quantified positive FOL formula $g$ is of the form $\exists x_1, \ldots, \exists x_k.\Phi(x_1, \ldots, x_k)$, where $\Phi$ is quantifier-free and does not contain any negation. For a positive instance $G$, a ME is the smallest subgraph $R$ induced by nodes in a tuple $\bar{a} = (a_1, \ldots, a_k)$, such that $\Phi(\bar{a})$ holds. Now, any supergraph of $R$ necessarily contains $\bar{a}$ and hence witnesses $\exists x_1, \ldots, \exists x_k.\Phi(x_1, \ldots, x_k)$, while any smaller subgraph violates $\exists x_1, \ldots, \exists x_k.\Phi(x_1, \ldots, x_k)$, as $\bar{a}$ is minimal by construction. Hence, $R$ is a PI. $\square$

Note that tasks based on the recognition of a topological motif (like the existence of a star) can indeed be cast as existentially quantified positive[4] formulas ($\exists xyzw.E(x,y) \wedge E(x,z) \wedge E(x,w) \wedge x \neq y \neq z \neq w$), qualifying MEs as ideal targets for those tasks. Motif-based tasks are indeed useful in a large class of practically relevant scenarios (Sushko et al., 2012; Jin et al., 2020; Chen et al., 2022; Wong et al., 2024), and have been a central assumption in many works on GNN explainability (Ying et al., 2019; Miao et al., 2022a; Wu et al., 2022). Theorem 4.2 theoretically supports using SE-GNNs for these tasks, and shows they optimize for minimally sufficient explanations (PIs). However, global properties such as long-range dependencies (Gravina et al., 2022) or classifiers like $\forall x \exists y E(x,y)$, cannot be expressed by purely existential statements. In such scenarios, PIs can be more informative than MEs, as we show next.

---

[4] Note that the positive fragment of FOL admits inequalities like $(x \neq y)$ as positive atoms (Kuperberg, 2023).

## 4.2. MEs are Not More Informative than PIs

Although Section 4.1 shows that MEs match PIs for positive existential tasks, real-world properties like counting cannot be expressed by existential formulas. Here, we show that beyond positive existential tasks, MEs can be less informative than PIs (Remark 3.3).

**Proposition 4.3.** *Let $g$ be a classifier and $y$ a given label. Let $\Omega_g^{(y)}$ be the set of all the finite graphs (potentially with a given bounded size) with predicted label $y$. Then,*

$$\bigcup_{G \in \Omega_g^{(y)}} \mathrm{ME}(g(G)) \ \subseteq \ \bigcup_{G \in \Omega_g^{(y)}} \mathrm{PI}(g(G)) \qquad (4)$$

This result shows that when considering the union of all finite graphs, PIs subsume MEs. Hence, if two classifiers share all PIs then they necessarily share all MEs as well, meaning that MEs do not provide more information about the underlying classifier than PIs. Conversely, we show that there exist cases where PIs provide strictly more information about the underlying classifier than MEs.

**Theorem 4.4.** *There exist two distinct classifiers $g$ and $g'$ and a label $y$ such that:*

1. *For all graphs $G$ s.t. $g(G) = g'(G) = y$, we have*

$$\mathrm{ME}\big(g(G)\big) \ = \ \mathrm{ME}\big(g'(G)\big). \qquad (5)$$

2. *There exists at least one graph $G^*$ with $g(G^*) = g'(G^*) = y$ such that*

$$\mathrm{PI}\big(g(G^*)\big) \ \neq \ \mathrm{PI}\big(g'(G^*)\big). \qquad (6)$$

Theorem 4.4 shows that PIs can overcome MEs' limits in certain tasks. For example, Theorem 3.4's classifiers yield identical MEs but differing PIs (see Fig. 1).

To summarize, Theorem 4.2 justifies SE-GNNs' performance on many benchmarks, as their explanations can inherit desirable properties of PIs. However, Theorem 3.4 and Theorem 4.4 show that MEs may not be informative for certain tasks, motivating an extension of SE-GNNs that preserves their performance on positive existential tasks but adaptively aids them in other tasks. We will exploit this observation for our proposed method in Section 6.

## 5. Minimal Explanations can be Unfaithful

A widespread approach to estimating the trustworthiness of an explanation is faithfulness (Yuan et al., 2022; Amara et al., 2022; Agarwal et al., 2023; Longa et al., 2024), which consists in checking whether the explanation contains all and only the elements responsible for the prediction. Clearly, unfaithful explanations fail to convey actionable information about what the model is doing to build its predictions (Agarwal et al., 2024). This section reviews a general notion of faithfulness (Azzolin et al., 2025) in Definition 5.1 and shows that faithfulness, MEs (Section 3), and PIs (Section 4) can overlap – in some restricted cases – but are generally misaligned.

Intuitively, faithfulness metrics assess how much the prediction changes when perturbing either the complement – referred to as *sufficiency* of the explanation – or the explanation itself – referred to as *necessity* of the explanation. Perturbations are sampled from a distribution of allowed modifications, which typically include edge and node removals. Then, an explanation is said to be faithful when it is both sufficient and necessary.

**Definition 5.1** (*Faithfulness*). Let $R \subseteq G$ be an explanation for $g(G)$ and $C = G \setminus R$ its complement. Let $p_R$ be a distribution over perturbations to $C$, and $p_C$ be a distribution over perturbations to $R$. Also, let $\Delta(G, G') = \mathbb{1}\{g(G) \neq g(G')\}$ indicate a change in the predicted label. Then, Suf and Nec compute respectively the ***degree of sufficiency*** and ***degree of necessity*** of an explanation $R$ for a decision $g(G)$ as:

$$\mathsf{Suf}(R) = \exp(-\mathbb{E}_{G' \sim p_R}[\Delta(G, G')]), \qquad (7)$$
$$\mathsf{Nec}(R) = 1 - \exp(-\mathbb{E}_{G' \sim p_C}[\Delta(G, G')]). \qquad (8)$$

The ***degree of faithfulness*** $\mathsf{Faith}(R)$ is then the harmonic mean of Suf and Nec.

The (negated) exponential normalization ensures that Suf (resp. Nec) increases for more sufficient (resp. necessary) explanations. Note that both Suf and Nec need to be non-zero for a Faith score above zero. According to this definition, the degree of necessity will be high when many perturbations of $R$ lead to a prediction change and is thus 1 when all of them do so. Suf is analogous.

As the next theorem shows, MEs can surprisingly fail in being faithful even for trivial tasks.

**Theorem 5.2.** *There exist tasks where Minimal Explanations achieve a zero degree of faithfulness.*

*Proof.* Let $g = \exists x \exists y. E(x, y)$ and $G$ be a positive instance with more than one edge. Then, a Minimal Explanation $R$ consisting of a single edge (see Fig. 1) achieves a $\mathsf{Nec}(R)$ value of zero, as any perturbations to $R$ will leave $G \setminus R$ unchanged. Hence, $g$ remains satisfied, meaning that $\mathsf{Nec}(R) = 0$ and thus $\mathsf{Faith}(R) = 0$. □

Note that $\exists x \exists y. E(x, y)$ achieves non-zero $\mathsf{Nec}(R)$ iff $R$ contains every edge, whereas $\mathsf{Suf}(R) = 1$ is guaranteed by the presence of a single edge. These observations indeed generalize, as shown next.

**Proposition 5.3.** *An explanation $R \subseteq G$ for $g(G)$ has a maximal $\mathsf{Suf}(R)$ score if and only if there exists a PI explanation $M \subseteq R$ for $g(G)$.*

**Proposition 5.4.** *An explanation $R \subseteq G$ for $g(G)$ can have a non-zero $\mathsf{Nec}(R)$ score only if it intersects every PI explanation for $g(G)$.*

Note that Proposition 5.3 (along with Theorem 4.2) show that MEs are indeed maximally sufficient for existentially quantified positive FOL formulas. However, this might not be enough to ensure a non-zero Faith score, as shown in Proposition 5.4. In general, MEs highlight a minimal class-discriminative subgraph, whereas faithful explanations highlight all possible causes for the prediction, and the two notions are fundamentally misaligned.

# 6. Beyond (Purely) Topological SE-GNNs

Our analysis highlights that explanations extracted by SE-GNNs can be, in general, both ambiguous (Theorem 3.4) and unfaithful (Theorem 5.2), potentially limiting their usefulness for debugging (Teso et al., 2023; Bhattacharjee & von Luxburg, 2024; Fontanesi et al., 2024), scientific discovery (Wong et al., 2024), and Out-of-Distribution (OOD) generalization (Chen et al., 2022; Gui et al., 2023). Nonetheless, we also showed that for a specific class of tasks, precisely those based on the presence of motifs, MEs are actually an optimal target, in that they match PIs and are the minimal explanations that are also provably sufficient (Theorem 4.2 and Proposition 5.3).

As PI and faithful explanations can generally be more informative than MEs (Proposition 4.3 and Theorem 4.4), one might consider devising SE-GNNs that always optimize for them. We argue, however, that aiming for PI or faithful explanations can be suboptimal, as they can be large and complex in the general case. For example, a PI explanation for a positive instance of $\forall x \exists y. E(x, y)$ is an edge cover (Weisstein, 2025), which grows with the size of the instance. Similarly, the explanation with an optimal faithfulness score for $\exists x \exists y. E(x, y)$ is the whole graph. Furthermore, finding and checking such explanations is inherently intractable, as both involve checking the model's output on all possible subgraphs (Yuan et al., 2021). These observations embody the widely observed phenomenon that explanations can get as complex as the model itself (Rudin, 2019).

## 6.1. Dual-Channel GNNs

This motivates us to investigate ***Dual-Channel GNNs*** (DC-GNNs), a new family of SE-GNNs aiming to preserve the perks of MEs for motif-based tasks while freeing them from learning non-motif-based patterns, which would be inefficiently represented as a subgraph. To achieve this, DC-GNNs pair an SE-GNN-based *topological channel* – named Topo– with an *interpretable channel* – named Rule– providing succinct non-topological rules. To achieve optimal separation of concerns while striving for simplicity, we

jointly train the two channels and let DC-GNNs adaptively learn how to exploit them.

**Definition 6.1.** (DC-GNN) Let $g_1 : G \mapsto [0, 1]^{n_1}$ be a SE-GNN, $g_2 : G \mapsto [0, 1]^{n_2}$ an interpretable model of choice, and $\mathsf{aggr} : [0, 1]^{n_1 + n_2} \mapsto Y$ an aggregation function. A DC-GNN is defined as:

$$g(G) = \mathsf{aggr}\big(g_1(G), g_2(G)\big). \qquad (9)$$

While the second channel $g_2$ can be any interpretable model of choice, we will experimentally show that even a simple sparse linear model can bring advantages over a standard SE-GNN, leaving the investigation of a more complex architecture to future work. Therefore, in practice, we will experiment with

$$g_2(G) = \sigma\big(W \textstyle\sum_{u \in G} x_u\big), \qquad (10)$$

where $x_u \in \mathbb{R}^d$ is the node feature vector of $u$, $W \in \mathbb{R}^{n_2 \times d}$ the weights of the linear model, and $\sigma$ is an element-wise sigmoid activation function. Also, we will fix $n_1 = n_2$ equal to the number of classes of $Y$. In the experiments, we will promote a sparse $W$ via weight decay.

A key component of DC-GNNs is the $\mathsf{aggr}$ function joining the two channels. To preserve the interpretability of the overall model, $\mathsf{aggr}$ should act as a gateway for making the final predictions based on a simple combination of the two channels. To achieve this while ensuring differentiability, we considered an extension of Logic Explained Networks (LENs) (Barbiero et al., 2022), described below.

**Implementing** $\mathsf{aggr}$. LENs are small Multi-layer Perceptrons (MLP) taking as input a vector of activations in $[0, 1]$ (Barbiero et al., 2022). The first layer of the MLP is regularized to identify the most relevant inputs while penalizing irrelevant ones via an attention mechanism. To promote the network to achieve discrete activations scores without resorting to high-variance reparametrization tricks (Azzolin et al., 2022; Giannini et al., 2024), we propose augmenting LENs with a progressive temperature annealing of the input vector. We validate the effectiveness of this extension in preserving the semantics of each channel, together with a comparison to other baselines, in Appendix C.4.

# 7. Empirical Analysis

We empirically address the following research questions:

**Q1** Can DC-GNNs match or surpass plain SE-GNNs?

**Q2** Can DC-GNNs adaptively select the best channel for each task?

**Q3** Can DC-GNNs extract more focused explanations?

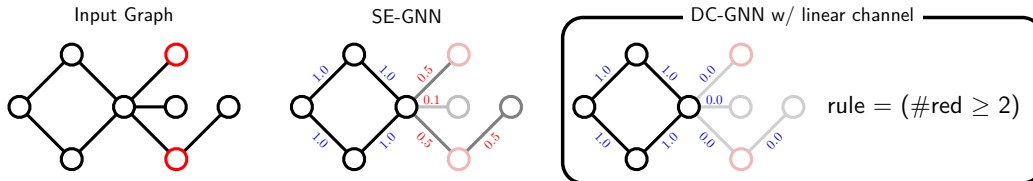

Figure 2: **Illustration of our Dual-Channel architecture** for a positive instance of `TopoFeature`, where positive instances contain a cycle and at least two red nodes. Numbers indicate edge relevance scores. SE-GNNs may fail to highlight precisely the elements relevant to the prediction, as the task involves non-topological patterns. DC-GNNs, instead, provide more focused topological explanations by offloading part of the prediction to interpretable rules (cf. Fig. 7 and Fig. 10).

Full details about the empirical analysis are in Appendix B. Our code is publicly available on GitHub[5].

**Architectures**. We test two representative SE-GNNs backbones from Table 1, namely GISST (Lin et al., 2020) and GSAT (Miao et al., 2022a), showing how adding an interpretable side channel can enhance them. We also propose a new SE-GNN named SMGNN, which replaces the graph-agnostic GISST's explanation extractor with that of GSAT. More details are available in Appendix B.

**Datasets**. We consider three synthetic and five real-world graph classification tasks. Synthetic datasets include GOODMotif (Gui et al., 2022), and two novel datasets. RedBlueNodes contains random graphs where each node is either red or blue, and the task is to predict which color is more frequent. Similarly, TopoFeature contains random graphs where each node is either red or uncolored, and the task is to predict whether the graph contains at least two red nodes *and* a cycle, which is randomly attached to the base graph. Both datasets contain two OOD splits where the number of total nodes is increased or the distribution of the base graph is changed. For GOODMotif we will use the original OOD splits (Gui et al., 2022). Real-world datasets include MUTAG (Debnath et al., 1991), BBBP (Morris et al., 2020), MNIST75sp (Knyazev et al., 2019), AIDS (Riesen & Bunke, 2008), and Graph-SST2 (Yuan et al., 2022).

**A1: DC-GNNs performs on par or better than plain SE-GNNs**. We list in Table 2 the results for synthetic datasets and in Table 3 those for real-world benchmarks. In the former, DC-GNNs always match or surpass the performance of plain SE-GNNs, and similarly, in the latter DC-GNNs perform competitively with SE-GNNs baselines, and in some cases can even surpass them all while keeping a comparable running time as SE-GNNs as shown in Appendix C.1. For example, DC-GNNs consistently surpass the baseline on AIDS and Graph-SST2 by relying on the non-relational interpretable model. These results are in line with previous literature highlighting that some tasks can be unsuitable for

testing topology-based explainable models (Azzolin et al., 2025), and that graph-based architectures can overfit the topology to their detriment (Bechler-Speicher et al., 2023). We present further experiments in Appendix C.2 and Appendix C.3, where we show the sensitivity of DC-GNNs to different hyper-parameter choices, and that DC-GNNs are also applicable to SE-GNNs beyond Table 1, respectively.

We further analyze their generalization abilities by reporting their performance on OOD splits in Table 4. Surprisingly, DC-GNNs significantly outperformed baselines on $OOD_1$ of GOODMotif but underperformed on $OOD_2$, likely due to the intrinsic instability of OOD performance in models trained without OOD regularization (Chen et al., 2022; Gui et al., 2023). For RedBlueNodes and TopoFeature instead, DC-GNNs exhibit substantial gains in seven cases out of eight, achieving perfect extrapolation on RedBlueNodes.

**A2: DC-GNNs can dependably select the appropriate channel(s) for the task**. Results in Table 2 clearly show that both DC-GNNs correctly identify the appropriate channel(s) for all tasks. In particular, they focus on the interpretable model for RedBlueNodes as the label can be predicted via a simple comparison of node features. For GOODMotif, they rely on the underlying SE-GNN only, as the label depends on a topological motif. For TopoFeature, instead, they combine both channels as the task can be decomposed into a motif-based sub-task and a comparison of node features, as expected from the dataset design. While the channel importance scores predicted by the aggr are continuous, we report a discretized version indicating only the channel(s) achieving a non-negligible ($\geq 0.1$) score to avoid clutter. Raw scores are reported in Table 8.

To measure the dependability of the channel selection mechanism of DC-GNNs, we perform an ablation study by setting to zero both channels independently. We report the resulting accuracy in the last two columns for every dataset in Table 2. Results show that when the model focuses on just a single channel, removing the other does not affect performance. Conversely, when the model finds it useful to mix information from both channels, removing either of them prevents

---

[5]https://github.com/steveazzolin/beyond-topo-segnns

Table 2: **DC-GNNs can reliably select the most appropriate channel based on the task.** The table shows the test accuracy, the selected channel, and the test accuracy when removing one of the two channels. GISST is excluded from this analysis as it cannot extract meaningful explanations for TopoFeature and GOODMotif.

| Model | RedBlueNodes | | | | TopoFeature | | | | GOODMotif | | | |
|---|---|---|---|---|---|---|---|---|---|---|---|---|
| | Acc | Channel | Acc w/o Topo | Acc w/o Rule | Acc | Channel | Acc w/o Topo | Acc w/o Rule | Acc | Channel | Acc w/o Topo | Acc w/o Rule |
| GIN | $99 \pm 01$ | - | - | - | $96 \pm 02$ | - | - | - | $93 \pm 00$ | - | - | - |
| GSAT | $99 \pm 01$ | - | - | - | $99 \pm 01$ | - | - | - | $93 \pm 00$ | - | - | - |
| DC-GSAT | $100 \pm 00$ | Rule | $100 \pm 01$ | $50 \pm 02$ | $100 \pm 00$ | Both | $50 \pm 00$ | $60 \pm 20$ | $93 \pm 01$ | Topo | $34 \pm 01$ | $93 \pm 00$ |
| SMGNN | $99 \pm 00$ | - | - | - | $95 \pm 01$ | - | - | - | $93 \pm 01$ | - | - | - |
| DC-SMGNN | $100 \pm 00$ | Rule | $99 \pm 01$ | $50 \pm 02$ | $100 \pm 00$ | Both | $50 \pm 00$ | $50 \pm 00$ | $93 \pm 00$ | Topo | $34 \pm 01$ | $93 \pm 00$ |

Table 3: **DC-GNNs performs on par or better than a plain SE-GNN.** The table shows the test accuracy and the selected channel for real-world experiments. Channel relevance with "*" means that DC-GNNs select the indicated channel in nine seeds over ten: For AIDS, that single seed uses the topological channel, as it can easily fit the task without incurring performance loss; For BBBP, instead, the DC-GNN tries to use both channels but incurs performance loss. "Mix" instead indicates that different seeds rely on different channels without incurring performance loss.

| Model | AIDS | | MUTAG | | BBBP | | Graph-SST2 | | MNIST75sp | |
|---|---|---|---|---|---|---|---|---|---|---|
| | F1 | Channel | Acc | Channel | AUC | Channel | Acc | Channel | Acc | Channel |
| GIN | $96 \pm 02$ | - | $81 \pm 02$ | - | $68 \pm 03$ | - | $88 \pm 01$ | - | $92 \pm 01$ | - |
| GISST | $97 \pm 02$ | - | $78 \pm 02$ | - | $66 \pm 03$ | - | $85 \pm 01$ | - | $91 \pm 01$ | - |
| DC-GISST | $99 \pm 02$ | Mix | $79 \pm 02$ | Topo | $65 \pm 02$ | Topo | $87 \pm 02$ | Mix | $91 \pm 01$ | Topo |
| GSAT | $97 \pm 02$ | - | $79 \pm 02$ | - | $66 \pm 02$ | - | $86 \pm 02$ | - | $94 \pm 01$ | - |
| DC-GSAT | $99 \pm 03$ | Rule | $79 \pm 02$ | Topo | $65 \pm 03$ | Topo* | $87 \pm 02$ | Rule | $94 \pm 02$ | Topo |
| SMGNN | $97 \pm 02$ | - | $79 \pm 02$ | - | $66 \pm 03$ | - | $86 \pm 01$ | - | $92 \pm 01$ | - |
| DC-SMGNN | $99 \pm 01$ | Rule* | $80 \pm 02$ | Topo | $65 \pm 05$ | Topo* | $87 \pm 01$ | Rule | $93 \pm 01$ | Topo |

Table 4: **DC-GNNs can generalize better to OOD than plain SE-GNN.** GISST is excluded from this analysis as it cannot extract meaningful explanations for TopoFeature and GOODMotif. Numbers indicate OOD accuracy.

| Model | RedBlueNodes | | TopoFeature | | Motif | |
|---|---|---|---|---|---|---|
| | $OOD_1$ | $OOD_2$ | $OOD_1$ | $OOD_2$ | $OOD_1$ | $OOD_2$ |
| GIN | $94 \pm 01$ | $87 \pm 05$ | $61 \pm 06$ | $61 \pm 04$ | $63 \pm 13$ | $64 \pm 03$ |
| GSAT | $98 \pm 01$ | $98 \pm 01$ | $81 \pm 08$ | $87 \pm 04$ | $67 \pm 12$ | $57 \pm 04$ |
| DC-GSAT | $100 \pm 00$ | $100 \pm 00$ | $87 \pm 13$ | $86 \pm 12$ | $72 \pm 13$ | $49 \pm 05$ |
| SMGNN | $97 \pm 01$ | $85 \pm 05$ | $55 \pm 02$ | $75 \pm 03$ | $65 \pm 06$ | $62 \pm 06$ |
| DC-SMGNN | $99 \pm 01$ | $100 \pm 00$ | $93 \pm 06$ | $98 \pm 01$ | $82 \pm 08$ | $43 \pm 04$ |

the model from making the correct prediction, as expected.

**A3: DC-GNNs can uncover high-quality rules and MEs.** When the linear model $g_2$ is sparse, it is possible to interpret the weights as inequality rules enabling a full understanding of the predictions, cf. Appendix B.3. In fact, by focusing on model weights with non-negligible magnitude, we can extract the expected ground truth rule for RedBlueNodes, whilst subgraph-based explanations fail to convey such a simple rule as shown in Appendix C.7. At the same time, for TopoFeature the interpretable model fires when the number of red nodes is at least two. This result, together with the fact that the DC-GNN is using both channels (see

Table 2), confirms that the two channels are cooperating as expected: the interpretable model fires when at least two red nodes are present, and the SE-GNN is left to recognize when the motif is present, achieving more focused and compact explanations as depicted in Fig. 2. The alignment of those formulas to the ground truth annotation process is also reflected in better extrapolation performance, as shown in Table 4. We further discuss in Appendix C.5 an additional experiment on AIDS where we match the result of Pluska et al. (2024), showing that a GNN can achieve a perfect score by learning a simple rule on the number of nodes. Those results highlight that better and more intuitive explanations can be obtained without relying on a subgraph of the input. We further provide a quantitative (Appendix C.6) and qualitative (Appendix C.7) analysis showing how DC-GNN can yield explanations that better reflect what the underlying SE-GNN is using for predictions.

## 8. Related Work

**Explaining GNNs**. SE-GNNs are a physiological response to the inherent limitations of *post-hoc* GNN explainers (Longa et al., 2024; Li et al., 2024). SE-GNNs usually rely on regularization terms and architectural biases – such as attention (Miao et al., 2022a;b; Lin et al., 2020; Serra &

Niepert, 2022; Wu et al., 2022; Chen et al., 2024), prototypes (Zhang et al., 2022; Ragno et al., 2022; Dai & Wang, 2021; 2022), or other techniques (Yu et al., 2020; 2022; Giunchiglia et al., 2022; Spinelli et al., 2023; Ferrini et al., 2024) – to encourage the explanation to be human interpretable. Our work aims at understanding the formal properties of their explanations, which have so far been neglected.

**Beyond subgraph explanations.** Pluska et al. (2024) and Köhler & Heindorf (2024) proposed to distill a trained GNN into an interpretable logic classifier. Their approach is however limited to *post-hoc* settings, and the extracted explanations are human-understandable only for simple datasets. Nonetheless, this represents a promising future direction for integrating a logic-based rule extractor as a side channel in our DC-GNNs framework. Müller et al. (2023) and Bechler-Speicher et al. (2024) introduced two novel interpretable-by-design GNNs that avoid extracting a subgraph explanation altogether, by distilling the GNN into a Decision Tree, or by modeling each feature independently via learnable shape functions, respectively. However, we claim that subgraph-based explanations are desirable for existential motif-based tasks, and thus, we strike for a middle ground between subgraph and non-subgraph-based explanations.

**Formal explainability**. While GNN explanations are commonly evaluated in terms of faithfulness (Agarwal et al., 2023; Christiansen et al., 2023; Azzolin et al., 2025), formal explainability has predominantly been studied for non-relational data, where it primarily focuses on PI explanations (Marques-Silva, 2023; Darwiche & Hirth, 2023; Wang et al., 2021). Important properties of PI explanations include sufficiency, minimality, and their connection to counterfactuals (Marques-Silva & Ignatiev, 2022). Simultaneously, PI explanations can be exponentially many (but can be summarized (Yu et al., 2023) for understanding) and are intractable to find and enumerate for general classifiers (Marques-Silva, 2023). Our work is the first to systematically investigate formal explainability for GNNs and elucidate the link between PI explanations and SE-GNNs.

## 9. Conclusion and Limitations

We have formally characterized the explanations given by sparsity- and Information Bottleneck-based Self-Explainable Graph Neural Networks (SE-GNNs) as Minimal Explanations (MEs). We identified and analyzed MEs' relation to established notions of explanations, like Prime Implicant (PI) and faithful explanations. Our analysis revealed that MEs match PIs and achieve maximal sufficiency score for motif-based tasks. However, in general MEs can be uninformative, whereas faithful and PI explanations can be large and intractable to find. Motivated by this, we introduced Dual-Channel GNNs (DC-GNNs), a new class of SE-GNNs that adaptively provide either MEs, interpretable

rules, or their combination. We empirically validate DC-GNNs, confirming their promise.

**Limitations**: Our theoretical analysis is focused on Sparsity- and IB-based SE-GNNs, which are popular choices due to their effectiveness and ease of training. Alternative formulations optimize explanations for different objectives than minimality (Wu et al., 2022; Deng & Shen, 2024), and more work is needed to provide a unified formalization of this broader class of subgraph-based explanations. Nonetheless, we expect our theoretical findings to also hold for many *post-hoc* explainers, as they also optimize for the minimal subgraph explaining the prediction (Ying et al., 2019; Luo et al., 2020). Furthermore, despite showing that DC-GNNs with a simple linear model have several benefits, we believe that more advanced interpretable channels should be investigated. On this line, our baseline can serve as a yardstick for future developments of DC-GNNs.

## Impact Statement

Our analysis highlights the intrinsic limitations of explanations produced by a wide class of explainable-by-design GNNs, and in this sense serves as a warning for stakeholders from blindly trusting explanations produced by these models.

## Acknowledgments

Funded by the European Union. Views and opinions expressed are however those of the author(s) only and do not necessarily reflect those of the European Union or the European Health and Digital Executive Agency (HaDEA). Neither the European Union nor the granting authority can be held responsible for them. Grant Agreement no. 101120763 - TANGO. SM acknowledges the support of FWF and ANR project NanOX-ML (6728).

The authors are grateful to Manfred Jaeger for useful discussions and feedback on an earlier version of this manuscript.

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

# A. Proofs

## A.1. Proof of Theorem 3.2

**Preliminaries and Assumptions.** Recall that an SE-GNN $g$ is composed of a GNN-based classifier $f$ and an explanation extractor $q$. Given an instance $G$, the explanation extractor $q$ returns a subgraph $q(G) = R \subseteq G$, and the classifier provides a label $g(G) = f(q(G))$.

Note that analyzing the losses presented in Table 1 is challenging due to the potentially misleading minima induced by undesirable choices of $\lambda_1$ and $\lambda_2$. For example, choosing the explanation regularization weight $\lambda_1$ and $\lambda_2$ in Table 1 to be zero can trivially lead to correct predictions but with uninformative explanations. Equivalently, setting $\lambda_1$ and $\lambda_2$ too high yields models with very compact yet useless explanations, as the model may not converge to a satisfactory accuracy. Since our goal is to analyze the nature of explanations extracted by $q$, we assume that $f$ expresses the ground truth function. Also, we assume the SE-GNN to have perfect predictive accuracy, meaning that $f(q(G))$ always outputs the ground truth label for any input. Those two assumptions together allow us to focus only on the nature of the explanations extracted by $q$.

We also consider an SE-GNN with a hard explanation extractor. For sparsity-based losses (Eq. (11)), this amounts to assigning a score equal to 1 for edges in the explanation $R$, and equal to 0 for edges in the complement $G \setminus R$. For Information Bottleneck-based losses (Eq. (14)), instead, the explanation extractor assigns a score of 1 for edges in the explanation $R$, and equal to $r$ for edges in the complement $G \setminus R$, where $r$ is the hyper-parameter chosen as the uninformative baseline for training the model (Miao et al., 2022a). Therefore, an explanation $R$ is identified as the edge-induced subgraph where edges have a score $p_{uv} = 1$.

**Theorem 3.2.** *Let $g$ be an SE-GNN with a ground truth classifier $f$ (i.e., $f(G)$ always returns the true label for $G$), a hard explanation extractor $q$, and perfect predictive accuracy. Then, $g$ achieves minimal true risk (as indicated in Table 1) if and only if for any instance $G$, $q(G)$ provides Minimal Explanations for the predicted label $g(G)$.*

*Proof.* We proceed to prove the Theorem by analyzing two cases separately:

**Sparsity-based losses** Let us consider the following training objective of a prototypical sparsity-based SE-GNN, namely GISST (Lin et al., 2020). Here we focus just on the sparsification of edge-wise importance scores, and we discuss at the end how this applies to node feature-wise scores:

$$\min \ \mathcal{L}\big(f(q(G)), Y\big) + \lambda_1 \frac{1}{|E|} \sum_{(u,v) \in E} p_{uv} + \lambda_2 \frac{1}{|E|} \sum_{(u,v) \in E} p_{uv} \log(p_{uv}) + (1 - p_{uv}) \log(1 - p_{uv}) \tag{11}$$

Given that the importance scores $p_{uv}$ can only take values in $\{0, 1\}$, the last term in Eq. (11) equals to 0. Also, given that every edge outside of $q(G)$ has $p_{uv} = 0$ and every edge in $q(G)$ has $p_{uv} = 1$, we have that

$$\sum_{(u,v) \in E} p_{uv} = |q(G)| \tag{12}$$

Hence, the final minimization reduces to:

$$\min \ \mathcal{L}\big(f(q(G)), Y\big) + \lambda_1 \frac{|q(G)|}{|E|} \tag{13}$$

**Minimal True Risk for SE-GNN $\Rightarrow$ Minimal Explanations**
Given that $f$ is the ground truth classifier, then due to perfect predictive accuracy for $f(q(G))$ we have that $f(q(G)) = f(G) = y^*$, where $y^*$ is the ground truth label for $G$. This implies that $\mathcal{L}(f(q(G)), Y)$ is minimal. Now, for the true risk to be minimal we must additionally have that $\lambda_1 |q(G)|/|E|$ in Eq. (13) is minimized as well. Note that $q(G)$ returns the explanation $R$. Hence, minimizing $\lambda_1 |q(G)|/|E|$ in Eq. (13) requires that we find the smallest $R \subseteq G$ such that $\mathcal{L}(f(q(G)), Y)$ is also minimal, hence $f(q(G)) = f(R) = y^*$. Also, note that $f(R) = f(q(R))$ by perfect predictive accuracy of $f(q(R))$ and $f$ being the ground truth classifier. Hence, we have that for an instance $G$, $R \subseteq G$ is the smallest subgraph such that $f(q(G)) = f(q(R))$. Hence, $R$ is a Minimal Explanation for $f(q(G))$.

**Minimal True Risk for SE-GNN $\Leftarrow$ Minimal Explanations**

We now show the other direction of the statement, i.e., if $q(G)$ provides Minimal Explanations then an SE-GNN $g$ (with ground truth graph classifier $f$ and perfect predictive accuracy) achieves minimal true risk. Since, $g$ achieves perfect predictive accuracy, we have that $\mathcal{L}(f(q(G)), Y)$ is minimal. Furthermore, by definition of Minimal Explanations and assumption of perfect predictive accuracy, we have that $R$ is the smallest subgraph such that $f(q(G)) = f(q(R)) = y^*$. Hence, $\lambda_1 |q(G)|/|E|$ can not be further minimized in Eq. (13).

The same argument applies verbatim when adding the sparsification of (discrete) node feature explanations to Eq. (11), as prescribed in Lin et al. (2020).

**Information Bottleneck-based losses**  Let us consider the following training objective of a prototypical stochasticity injection-based SE-GNN, namely GSAT (Miao et al., 2022a). The same holds for other models like LRI (Miao et al., 2022b), GMT (Chen et al., 2024), and GIB[6] (Wu et al., 2020a):

$$\min \mathcal{L}\big(f(q(G)), Y\big) + \lambda_1 \sum_{(u,v) \in E} p_{uv} \log(\frac{p_{uv}}{r}) + (1 - p_{uv}) \log(\frac{1 - p_{uv}}{1 - r}) \tag{14}$$

By the hard explanation extractor assumption, $p_{uv} = 1$ when $(u, v) \in R$ and $r$ otherwise. Then, we can differentiate the contribution of each edge to the second term separately, depending on its importance score:
For edges where $p_{uv} = 1$:

$$\sum_{(u,v) \in E} p_{uv} \log(\frac{p_{uv}}{r}) + (1 - p_{uv}) \log(\frac{1 - p_{uv}}{1 - r}) = |q(G)| \log(\frac{1}{r}) \tag{15}$$

For edges where $p_{uv} = r$:

$$\sum_{(u,v) \in E} p_{uv} \log(\frac{p_{uv}}{r}) + (1 - p_{uv}) \log(\frac{1 - p_{uv}}{1 - r}) = 0 \tag{16}$$

We can then rewrite the minimization as follows:

$$\min \mathcal{L}\big(f(q(G)), Y\big) + \lambda_1 |q(G)| \log(\frac{1}{r}) \tag{17}$$

which optimizes the same objective as Eq. (13), as the terms not in common are constants. Thus, a similar argument to that of sparsity-based losses follows.

$\square$

### A.2. Proof of Theorem 4.2

**Theorem 4.2.** *Given a classifier $g$ expressible as a purely existentially quantified positive first-order logic formula and a positive instance $G$ of any size, then a Minimal Explanation for $g(G)$ is also a Prime Implicant explanation for $g(G)$.*

*Proof.* Let $g$ be a classifier that can be expressed as a boolean FOL formula of the form

$$\exists x_1 \exists x_2 \ldots \exists x_k \ \Phi(x_1, x_2, \ldots, x_k), \tag{18}$$

where $\Phi$ is positive and quantifier-free. A positive instance $G$ for $g$ is an instance such that $G \models g$. Now, let $R \subseteq G$ be a Minimal Explanation for $g(G)$, i.e, for $G \models \Phi$. We must show that $R$ is also a PI explanation.

We now show that $R$ satisfies conditions (1), (2) and (3) for PI Explanation as given in Definition 4.1

1. By definition of Minimal Explanation, we already have $R \subseteq G$. Hence, condition (1) is satisfied.

2. Since $R$ is a Minimal Explanation, we have that $R \models g$. Also, $g$ is purely existential and positive, hence there are specific elements $a_1, \ldots, a_k \in R$ witnessing $g$; that is, $R \models \Phi(a_1, \ldots, a_k)$. Now if $\Gamma$ is a subgraph of $G$ such that $R \subseteq \Gamma$, then all $a_i$ and relations containing $a_i$ remain in $\Gamma$. Hence $\Gamma$ also satisfies $g$ and therefore $\Gamma \models g$. This shows that every superset $\Gamma$ of $R$ inside $G$ satisfies $g$, satisfying the condition (2) in the PI explanation definition.

---

[6]Even though it is not designed for interpretability, as it predicts importance scores at every layer making the resulting explanatory subgraph not intelligible, here we also include GIB as a reference as it shares the same training objective.

3. Finally, we now show that $R$ is minimal. Note that since $R$ is a Minimal Explanation, there exists no $|R'| \leq |R|$, such that $R' \models g$. In particular, if there was a $R' \subset R$ such that $R' \models g$, that would contradict the minimality condition for a Minimal Explanation. Consequently, no such $R'$ can serve as a smaller PI explanation. This ensures condition (3).

$\square$

**Remark:** There exist classifiers that are not existentially quantified but Minimal Explanations still equal PI explanations. For instance, $\exists^{=1}x.Red(x)$ can not be expressed as a purely existentially quantified first-order logic formula for which Minimal Explanations still equal PI explanations.

### A.3. Proof of Proposition 4.3

**Proposition 4.3.** *Let $g$ be a classifier and $y$ a given label. Let $\Omega_g^{(y)}$ be the set of all the finite graphs (potentially with a given bounded size) with predicted label $y$. Then,*

$$\bigcup_{G \in \Omega_g^{(y)}} \mathrm{ME}(g(G)) \subseteq \bigcup_{G \in \Omega_g^{(y)}} \mathrm{PI}(g(G)) \tag{4}$$

*Proof.* Any Minimal Explanation $R$ for $g(G)$ is also an instance with a label $g(R) = g(G) = y$. We now show that a Minimal Explanation for $g(R)$ is also a PI explanation for $g(R)$, and hence it is contained in $\bigcup_{G \in \Omega_g^{(y)}} \mathrm{PI}(g(G))$. For any $R' \subset R$, we have that $g(R') \neq g(G)$ (as $R$ is a Minimal Explanation for $g(G)$), and hence $g(R') \neq g(R)$. Furthermore, any extension of $R$ (within $R$) does not lead to prediction change, vacuously. Hence, $R$ is a PI explanation for $g(R) = g(G) = y$. $\square$

### A.4. Proof of Theorem 4.4

**Theorem 4.4.** *There exist two distinct classifiers $g$ and $g'$ and a label $y$ such that:*

1. *For all graphs $G$ s.t. $g(G) = g'(G) = y$, we have*

$$\mathrm{ME}\big(g(G)\big) = \mathrm{ME}\big(g'(G)\big). \tag{5}$$

2. *There exists at least one graph $G^*$ with $g(G^*) = g'(G^*) = y$ such that*

$$\mathrm{PI}\big(g(G^*)\big) \neq \mathrm{PI}\big(g'(G^*)\big). \tag{6}$$

*Proof.* We continue with $g$ and $g'$ as given in the proof of Theorem 3.4, i.e., $g = \exists x \exists y. E(x, y)$ and $g' = \forall x \exists y. E(x, y)$. As shown in Theorem 3.4, condition 1 is true for $g$ and $g'$. Now, for any positively labeled graph $G^*$, $\mathrm{PI}(g(G^*))$ is the set of edges in $G^*$, whereas $\mathrm{PI}(g'(G^*))$ is the set of edge covers. As shown in Fig. 1, there exists a graph (say a triangle) such that the set of edge covers is different from the set of edges. $\square$

### A.5. Proof of Proposition 5.3

**Proposition 5.3.** *An explanation $R \subseteq G$ for $g(G)$ has a maximal $\mathsf{Suf}(R)$ score if and only if there exists a PI explanation $M \subseteq R$ for $g(G)$.*

*Proof.* An explanation $R$ is maximally sufficient if all possible edge deletions in $G \setminus R$ leave the predicted label $g(G')$ equal to $g(G)$, where $G'$ are the possible graphs obtained after perturbing $G \setminus R$. Equivalently, every possible extension $G'$ of $R$ in $G$ preserves the label. This is true if and only if $R$ is a PI explanation or there exists a subgraph $M \subset R$, which is a PI. $\square$

### A.6. Proof of Proposition 5.4

**Proposition 5.4.** *An explanation $R \subseteq G$ for $g(G)$ can have a non-zero $\mathsf{Nec}(R)$ score only if it intersects every PI explanation for $g(G)$.*

*Proof.* An explanation $R$ has zero necessity score if all possible edge deletions in $R$ leave the predicted label $g(G')$ equal to $g(G)$, where $G' \subseteq G$ are the possible subgraphs obtained after perturbations in $R$. Assume to the contrary that $R$ does not

intersect a PI explanation $M$, and has a non-zero necessity score. Then, there exists a graph $G'$ obtained by perturbing $R$ such that $g(G) \neq g(G')$. But note that $M \subseteq G'$ and $M$ is a PI by assumption, hence $g(G)$ must be equal to $g(G')$, leading to a contradiction. Therefore, $R$ must intersect all prime implicants to have a non-zero necessity score. □

## B. Implementation Details

### B.1. Datasets

In this study, we experimented on nine graph classification datasets commonly used for evaluating SE-GNNs. Among those, we also proposed two novel synthetic datasets to show some limitations of existing SE-GNNs. More details regarding each dataset follow:

- `RedBlueNodes` (ours). Nodes are colored with a one-hot encoding of either red or blue. The task is to predict whether the number of red nodes is larger or equal to the number of blue ones. The topology is randomly generated from a Barabási-Albert distribution (Barabási & Albert, 1999). Each graph contains a number of total nodes in the range $[10, 100]$. We also generate two OOD splits, where respectively either the number of total nodes is increased to $250$ ($\text{OOD}_1$), or where the distribution of the base graph is switched to an Erdos-Rényi distribution ($\text{OOD}_2$) (Erdos et al., 1959).

- `TopoFeature` (ours). Nodes are either uncolored or marked with a red color represented as one-hot encoding. The task is to predict whether the graph contains a certain motif together with at least two nodes. The base graph is randomly generated from a Barabási-Albert distribution (Barabási & Albert, 1999). Each graph contains a number of total nodes in the range $[8, 80]$. We also generate two OOD splits, where respectively either the number of total nodes is increased to $250$ ($\text{OOD}_1$), or where the distribution of the base graph is switched to an Erdos-Rényi distribution ($\text{OOD}_2$) (Erdos et al., 1959)

- `GOODMotif` (Gui et al., 2022) is a three-classes synthetic dataset for graph classification where each graph consists of a basis and a special motif, randomly connected. The basis can be a ladder, a tree (or a path), or a wheel. The motifs are a house (class 0), a five-node cycle (class 1), or a crane (class 2). The dataset also comes with two OOD splits, where the distribution of the basis changes, whereas the motif remains fixed (Gui et al., 2022). In our work, we refer to the OOD validation split of Gui et al. (2022) as $\text{OOD}_1$, while to the OOD test split as $\text{OOD}_2$.

- `MUTAG` (Debnath et al., 1991) is a molecular property prediction dataset, where each molecule is annotated based on its mutagenic effect. The nodes represent atoms and the edges represent chemical bonds.

- `BBBP` (Wu et al., 2018) is a dataset derived from a study on modeling and predicting barrier permeability (Martins et al., 2012).

- `AIDS` (Riesen & Bunke, 2008) contains chemical compounds annotated with binary labels based on their activity against HIV. Node feature vectors are one-hot encodings of the atom type.

- `AIDSC1` (ours) is an extension of `AIDS` where we concatenate the value $1.0$ to the feature vector of each node.

- `MNIST75sp` (Knyazev et al., 2019) converts the image-based digits inside a graph by applying a super pixelation algorithm. Nodes are then composed of superpixels, while edges follow the spatial connectivity of those superpixels.

- `Graph-SST2` is a sentiment analysis dataset based on the NLP task of sentiment analysis, adapted from the work of Yuan et al. (2022). The primary task is a binary classification to predict the sentiment of each sentence.

### B.2. Architectures

**SE-GNNs** The SE-GNNs considered in this study are composed of an explanation extractor $q$ and a classifier $f$. The explanation extractor is responsible for predicting edge (or equivalently node) relevance scores $p_{uv} \in [0, 1]$, which indicate the relative importance of that edge. Scores are trained to saturate either to $1$ or to a predetermined value that is considered as the uninformative baseline. For IB-based losses, this value corresponds to the parameter $r$ (Miao et al., 2022a;b), whereas for Sparsity based it equals $0$ (Lin et al., 2020; Yu et al., 2020). The classifier then takes as input the explanation and predicts the final label. Generally, both the explanation extractor and the classifier are implemented as GNNs, and relevance scores are predicted by a small neural network over an aggregated representation of the edge, usually represented

as the concatenation of the node representations of the incident nodes. A notable exception is `GISST`, using a *shallow* explanation extractor directly on raw features. Both explanation extractors and classifiers can then be augmented with other components to enhance their expressivity or their training stability like virtual nodes (Wu et al., 2022; Azzolin et al., 2025) and normalization layers (Ioffe, 2015). Then, SE-GNNs may also differ in their training objectives, as shown in Table 1, or the type of data they are applied to (Miao et al., 2022b).

We resorted to the codebase of Gui et al. (2022) for implementing `GSAT`, which contains the original implementation tuned with the recommended hyperparameters. `GISST` is implemented following the codebase of Christiansen et al. (2023). When reproducing the original results was not possible, we manually tuned the parameters to achieve the best downstream accuracies. We use the same explanation extractor for every model, implemented as a `GIN` (Xu et al., 2018), and adopt Batch Normalization (Ioffe, 2015) when the model does not achieve satisfactory results otherwise. Following Azzolin et al. (2025), we adopt the *explanation readout* mitigation in the final global readout of the classifier, so to push the final prediction to better adhere to the edge relevance scores. This is implemented as a simple weighted sum of final node embeddings, where the weights are the average importance score for each incident edge to that node. The only exceptions are `GOODMotif`, `BBBP`, and `Graph-SST2`, where we use a simple mean aggregator as the final readout to match the results of original papers.

**Model hyper-parameter.** We set the weight of the explanation regularization as follows: For `GISST`, we weight all regularization by $0.01$ in the final loss; For `SMGNN`, we set $1.0$ and $0.8$ the $L_1$ and entropy regularization respectively; For `GSAT`, we set the value of $r$ to $0.7$ for `GOODMotif`, `MNIST75sp`, `Graph-SST2`, and `BBBP`, to $0.5$ for `TopoFeature`, `AIDS`, `AIDSC1`, and `MUTAG`, and to $0.3$ for `RedBlueNodes`. Also, for `GSAT` we set the decay of $r$ is set every $10$ step for every dataset, except for `Graph-SST2` and `GOODMotif` where it is set to $20$. Then, the parameter $\lambda$ regulating the weight of the regularization is set to $0.001$ for all experiments with `SMGNN`, while to $1$ for `GSAT` on every dataset except for `RedBlueNodes`.

For each model, we set the hidden dimension of GNN layers to be $64$ for `MUTAG`, $300$ for `GOODMotif`, `BBBP`, and `Graph-SST2`, and $100$ otherwise. Similarly, we use a dropout value of $0.5$ for `GOODMotif` and `Graph-SST2`, of $0.3$ for `MNIST75sp`, `MUTAG`, and `BBBP`, and of $0.0$ otherwise.

**SMGNN.** The explanation extractor $q$ of `GISST` is implemented as a simple MLP over input features, meaning the explanation does not depend on topological information. To overcome this limitation, we propose a simple augmentation named Simple Modular GNN (`SMGNN`). `SMGNN` adopts the same explanation extractor as `GSAT`, which is however trained with the `GISST`'s sparsification loss after an initial warmup of $10$ epochs. Therefore, `SMGNN` adopts a specular architecture as `GSAT`, but with the sparsification loss of `GISST`. Similarly to `GSAT`, unless otherwise specified, the classifier backbone is shared with that of the explanation extractor, and composed of $5$ layers for `BBBP`, and $3$ layers for all the other datasets.

**Dual-Channel GNNs** Dual-Channel GNNs are implemented as a reference SE-GNN of choice, whose architecture remains unchanged, and a linear classifier taking as input a global sum readout of node input features. Both models have an output cardinality equal to the number of output classes, with a sigmoid activation function. Then, the outputs of the two models are concatenated and fed to our `(B) LENs`, described in Appendix C.4. Following Barbiero et al. (2022), we use an additional fixed temperature parameter with a value of $0.6$ to promote sharp attention scores of the underlying `LENs`. Also, the number of layers of `(B) LENs` and `LENs` are fixed to $3$ including input and output layer, and the hidden size is set to $350$ for `MNIST75sp`, $64$ for `MUTAG` and `BBBP`, $30$ for `GOODMotif`, and $20$ otherwise. The input layer of `(B) LENs` and `LENs` does not use the *bias* parameter. We adopt weight decay regularization to promote the sparsity of the linear model. For the additional experiment on `AIDSC1` (see Appendix C.5), a more stringent $L_1$ sparsification is applied.

### B.3. Extracting rules from linear classifiers

Although linear classifiers do not explicitly generate rules, their weights can be interpreted as inequality-based rules. For this interpretation to be meaningful, the model should remain simple, with sparse weights that promote clarity. In this section, we review how this can be achieved.

A (binary) linear classifier makes predictions based on a linear combination of input features, as follows:

$$y = w^T \mathbf{x} + b \tag{19}$$

where $\mathbf{x} \in \mathbb{R}^d$ is a $d$-dimensional input vector , and $w \in \mathbb{R}^d$ and $b \in \mathbb{R}$ the learned parameters. The decision boundary of the classifier corresponds to the hyperplane $w^T \mathbf{x} + b = 0$, and the classification is then based on the sign of $y$. We will

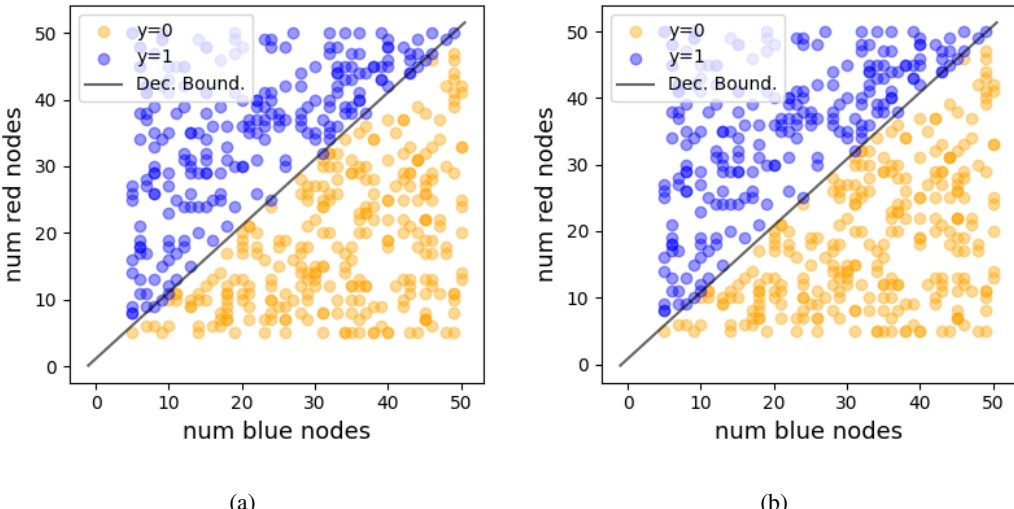

(a)                                      (b)

Figure 3: Decision boundary of the linear classifier of `DC-SMGNN` for `RedBlueNodes` over the validation split (random seed 1 and 2). The plot is in 2D since `RedBlueNodes` contains only red or blue nodes.

consider a classifier predicting the positive class when

$$w^T \mathbf{x} + b \geq 0. \tag{20}$$

By unpacking the dot product, Eq. (20) corresponds to a weighted summation of input features, where weights correspond to the model's weights $w$:

$$w_1 x_1 + \cdots + w_d x_d + b \geq 0. \tag{21}$$

For ease of understanding, let us commit to the specific example of `RedBlueNodes`. There, the linear classifier in the side channel of DC-GNNs takes as input the sum of node features for each graph. Therefore, our input vector will be $\mathbf{x} = [x_r, x_b, x_u]$, where $x_r, x_b, x_u$ indicates the number of red, blue, and uncolored nodes, and a positive prediction is made when

$$w_r x_r + w_b x_b + w_u x_u + b \geq 0. \tag{22}$$

If the model is trained correctly, and the training regime promotes enough sparsity – e.g. via weight decay or $L_1$ sparsification – we can expect to have $w_u \sim 0$ as there are no uncolored nodes and thus feature $x_u$ carries no information, and $b \sim 0$. Then, we can rewrite Eq. (22) as

$$w_r x_r \geq -w_b x_b \tag{23}$$

Then, $w_r = -w_b$ is a configuration of parameters that allows to perfectly solve the task, yielding the formula

$$x_r \geq x_b \tag{24}$$

We show in Fig. 3 that indeed our DC-GNNs learned such configuration of parameters by illustrating the decision boundary for the first two random seeds over the validation set. The figure confirms that the decision boundary is separating graphs based on the prevalence of red or blue nodes.

We plot in Fig. 4 a similar figure for `TopoFeature`, where the decision boundary intersects the y-axis, on average over 10 seeds, at the value of 1.23. By inspecting the model's weights, non-red nodes are assigned a sensibly lower importance magnitude (at least $10^2$ lower). Therefore, to convey a compact formula, we keep only the contribution of $x_r$, resulting on average in the final formula $x_r \geq 1.23$, which fires when at least two red nodes are present.

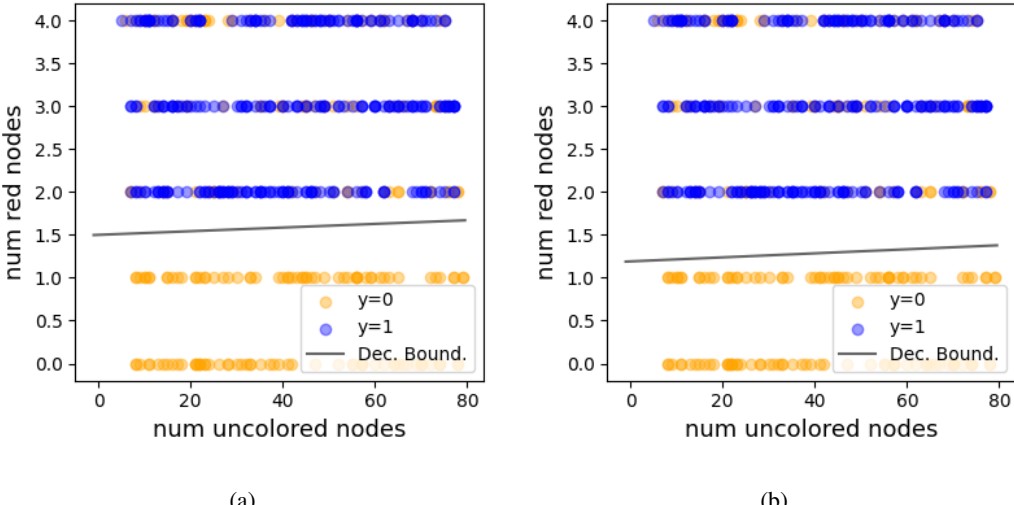

(a)                                    (b)

Figure 4: Decision boundary of the linear classifier of `DC-SMGNN` for `TopoFeature` over the validation split (random seed 1 and 2). For seed 1, $w_r = 0.89$, $w_b = 4e^{-42}$, $w_u = -0.001$, and $b = -1.3$. For seed 2, $w_r = 0.92$, $w_b = -1e^{-33}$, $w_u = -0.002$, and $b = -1.09$. Since $x_b$ equals 0 as there are no blue nodes in the dataset, we drop its visualization collapsing the plot to 2D. When providing the final interpretable formula, we drop $x_b$ and $x_u$ due to their sensible lower magnitude. More aggressive sparsification can be applied to the training of the linear model to promote an even lower $w_u$.

### B.4. Training and evaluation.

Every model is trained for the same 10 random splits, and the optimization protocol is fixed across all experiments following previous work (Miao et al., 2022a) and using the Adam optimizer (Kingma & Ba, 2015). Also, for experiments with Dual-Channel GNN, we fix an initial warmup of 20 epochs where the two channels are trained independently to output the ground truth label. After this warmup, only the overall model is trained altogether. The total number of epochs is fixed to 100 for every dataset except for `Graph-SST2` where it is set to 200.

For experiments on `Graph-SST2` we forced the classifier of any SE-GNNs to have a single GNN layer and a final linear layer mapping the graph embedding to the output. The parameters of the classifier are then different from those of the explanation extractor and trained jointly.

When training `SMGNN`, to avoid gradient cancellation due to relevance scores approaching exact zero, we use a simple heuristic to push the scores higher when their average value in the batch is below $2e^{-9}$. This is implemented by adding back to the loss of the batch the negated mean scaled by 0.1. This is similar to value clapping used by `GISST`, but we found it to yield better empirical performances.

### B.5. Software and hardware

Our implementation is done using PyTorch 2.4.1 (Paszke et al., 2017) and PyG 2.4.0 (Fey & Lenssen, 2019). Experiments are run on two different Linux machines, with CUDA 12.6 and a single NVIDIA GeForce RTX 4090, or with CUDA 12.0 and a single NVIDIA TITAN V.

## C. Additional Experiments

### C.1. Running time analysis

Table 5 presents the running time for two instances of SE-GNNs and DC-GNNs on real-world datasets, showing that the proposed architecture is not adding any significant computational overhead to the base architecture.

Table 5: Average running time in seconds of SE-GNNs and DC-GNNs for two real-world datasets.

| Model | Graph-SST2 | BBBP |
|---|---|---|
| GSAT | $4.78 \pm 0.23$ | $0.55 \pm 0.18$ |
| DC-GSAT | $4.50 \pm 0.32$ | $0.58 \pm 0.18$ |
| SMGNN | $6.51 \pm 0.37$ | $0.53 \pm 0.18$ |
| DC-SMGNN | $6.32 \pm 0.50$ | $0.71 \pm 0.22$ |

## C.2. Hyper-parameter ablation study

Table 6 presents an ablation study on the GOODMotif dataset to investigate the robustness of our results to the choice of hyper-parameter. Overall, results are stable across the hyper-parameter choice, except for a small fluctuation when choosing $r = 0.5$ as the parameter of GSAT. However, for $r \geq 0.7$, performance stabilizes.

Table 6: Ablation study on the hyper-parameter choice for the GOODMotif dataset. The values underlined represent chosen values for the main experiments. Results indicate test accuracy.

| Hyper-parameter | Value | DC-GSAT | DC-SMGNN |
|---|---|---|---|
| (B) LENs hidden size | 15 | $92 \pm 01$ | $93 \pm 01$ |
| | 30 | $93 \pm 01$ | $93 \pm 01$ |
| | 64 | $92 \pm 01$ | $93 \pm 01$ |
| (B) LENs num layers | 2 | $93 \pm 01$ | $93 \pm 01$ |
| | 3 | $93 \pm 01$ | $93 \pm 01$ |
| | 4 | $93 \pm 01$ | $93 \pm 01$ |
| (B) LENs sigmoid temperature | 0.1 | $93 \pm 01$ | $93 \pm 01$ |
| | 0.3 | $93 \pm 01$ | $93 \pm 01$ |
| | 0.5 | $93 \pm 01$ | $93 \pm 01$ |
| GSAT's $r$ | 0.5 | $90 \pm 02$ | - |
| | 0.7 | $93 \pm 01$ | - |
| | 0.9 | $93 \pm 01$ | - |

## C.3. Testing DC-GNNs on DIR (Wu et al., 2022)

Following the recommendation of an anonymous reviewer, we performed an additional experiment with DIR (Wu et al., 2022), which is a SE-GNNs not fitting the taxonomy of Table 1. While DIR may not output Minimal Explanations, our proposed Dual-Channel GNNs is a general framework that can be applied to any SE-GNNs. We report the results for augmenting DIR with our DC-SMGNN framework in Table 7. In running those experiments, we did not perform hyper-parameter tuning, thus final model accuracies and stability might be further improved by a more careful selection.

Table 7: Augmenting DIR with our proposed Dual-Channel GNNs framework.

| Model | AIDS | | MUTAG | | BBBP | | Graph-SST2 | | MNIST75sp | |
|---|---|---|---|---|---|---|---|---|---|---|
| | F1 | Channel | Acc | Channel | AUC | Channel | Acc | Channel | Acc | Channel |
| DIR | $96 \pm 03$ | - | $78 \pm 02$ | - | $64 \pm 02$ | - | $83 \pm 01$ | - | $83 \pm 02$ | - |
| DC-DIR | $96 \pm 03$ | Rule | $78 \pm 04$ | Mix | $65 \pm 02$ | Topo* | $83 \pm 01$ | Mix | $82 \pm 02$ | Topo |

## C.4. Ablation study on how to choose aggr

Our implementation of aggr relies on LENs to combine the two channels in an interpretable manner. LENs are however found to be susceptible to leakage (Azzolin et al., 2022), that is they can exploit information beyond those encoded in the activated class. Leakage hinders the semantics of input activations, thus comprising the interpretability of the prediction (Margeloiu et al., 2021; Havasi et al., 2022). Popular remedies include binarizing the input activations (Margeloiu et al., 2021; Azzolin et al., 2022; Giannini et al., 2024) so as to force the hidden layers of LENs to focus on the activations themselves, rather than their confidence.

Table 8: Raw channel relevance scores for the experiments in Table 2. The "Channel" column reports, the relative importance computed for each channel, For binary classification tasks, the first entry corresponds to the SE-GNN channel, whereas the last is to the interpretable model. For multi-class tasks, the first $|Y|$ entries correspond to the importance for each of the $|Y|$ SE-GNNs outputs, whereas the last $|Y|$ to that of the linear model. For `GOODMotif`, in particular, $|Y| = 3$.

| Model | RedBlueNodes | | TopoFeature | | GOODMotif | |
|---|---|---|---|---|---|---|
| | Acc | Channel | Acc | Channel | Acc | Channel |
| `DC-GSAT` | $100 \pm 00$ | $[0.02, 0.98] \pm 0.05$ | $100 \pm 00$ | $[0.47, 0.53] \pm 0.27$ | $93 \pm 01$ | $[0.33, 0.36, 0.30, 4e^{-3}, 2e^{-4}, 1e^{-4}]$ $\pm[0.33, 0.36, 0.29, 1e^{-2}, 2e^{-4}, 1e^{-4}]$ |
| `DC-SMGNN` | $100 \pm 00$ | $[0.05, 0.95] \pm 0.10$ | $100 \pm 00$ | $[0, 31, 0, 69] \pm 0.11$ | $93 \pm 00$ | $[0.27, 0.19, 0.53, 1e^{-4}, 3e^{-5}, 1e^{-3}]$ $\pm[0.39, 0.29, 0.39, 0.0, 0.0, 0.0]$ |

Table 9: Ablation study for the choice of `aggr`. The reference architecture in use is `SMGNN` and the dataset used for the evaluation is `TopoFeature`. The "Channel" column reports, when `aggr` supports it, the relative importance computed for each channel, where the first entry corresponds to the SE-GNN channel, whereas the last to the interpretable model.

| aggr | TopoFeature | | |
|---|---|---|---|
| | Acc | Channel | Num red nodes $\geq$ |
| $\text{aggr}_{\text{Gödel}}$ | $96 \pm 03$ | Not supported | - |
| $\text{aggr}_{\text{Product}}$ | $96 \pm 04$ | Not supported | - |
| $\text{aggr}_{\text{Linear}}$ | $79 \pm 11$ | $[-1.22, 1.77] \pm [1.65, 0.92]$ | $0.24 \pm 1.01$ |
| $\text{aggr}_{\text{MLP}}$ | $99 \pm 02$ | Not supported | - |
| $\text{aggr}_{\text{LENs}}$ | $95 \pm 05$ | $[0.62, 0.38] \pm 0.42$ | $0.17 \pm 1.47$ |
| $\text{aggr}_{\text{LENs}}^{ST}$ | $65 \pm 07$ | $[0.01, 0.99] \pm 0.01$ | $0.70 \pm 0.96$ |
| $\text{aggr}_{\text{(B)LENs}}$ (ours) | $100 \pm 00$ | $[0.31, 0.69] \pm 0.11$ | $1.23 \pm 0.17$ |

In our work, we adopt a different strategy to avoid leakage in LENs and propose to anneal the temperature of the input activations so as to gradually approach a binary activation during training, while ensuring a smooth differentiation. Input activations are generally assumed to be continuous in $[0, 1]$, and usually generated by a sigmoid activation function (Barbiero et al., 2022). Therefore, our temperature annealing simply scales the raw activation before the sigmoid activation by a temperature parameter $\tau$, where $\tau$ is linearly annealed from a value of $1$ to $0.3$. The resulting model is indicated with the name of (Binary)LENs− (B)LENs in short.

In the following ablation study, we investigate different approaches for implementing `aggr`, showing that only `(B)LENs` reliably achieve satisfactory performances while preserving the semantics of each channel. The baselines for implementing `aggr` considered in our study are as follows:

- Logic combination based on T-norm fuzzy logics, like Gödel $\text{aggr}_{\text{Gödel}}(A, B) = min(A, B)$ and Product logic $\text{aggr}_{\text{Product}}(A, B) = A * B$ (Klement et al., 2013).

- Linear combination $\text{aggr}_{\text{Linear}}(A, B) = W[A||B]$, where $||$ concatenaMEs the two inputs.

- Multi-Layer Perceptron $\text{aggr}_{\text{MLP}}(A, B) = W_3(\sigma(W_2(\sigma W_1[A||B])))$.

- Logic Explained Network (LENs) $\text{aggr}_{\text{LENs}}(A, B) = \text{LENs}(A||B)$.

- Logic Explained Network (LENs) with discrete input $\text{aggr}_{\text{LENs}}^{ST}(A, B) = \text{LENs}(ST(A)||ST(B))$. The discreteness of the input activations is obtained using the Straight-Trough (ST) reparametrization (Jang et al., 2016; Azzolin et al., 2022; Giannini et al., 2024), which uses the hard discrete scores in the forward step of the network, while relying on their soft continuous version for backpropagation.

We provide in Table 9 the results for `DC-SMGNN` using different `aggr` choices over `TopoFeature`. In this dataset, the expected rule to be learned by the interpretable channel is *number of red nodes $\geq 2$*. Among the alternatives, $\text{aggr}_{\text{Gödel}}$, $\text{aggr}_{\text{Product}}$, and $\text{aggr}_{\text{MLP}}$ do not allow to understand how the channels are combined, resulting in unsuitable `aggr` functions for our purpose. $\text{aggr}_{\text{Linear}}$ and $\text{aggr}_{\text{LENs}}^{ST}$, instead, fail in solving the task. $\text{aggr}_{\text{LENs}}$, however, achieve both a satisfactory accuracy and an interpretable combination of channels. Nonetheless, the presence of leakage hinders a full understanding of the interpretable rule, as the hidden layers of the LENs are allowed to exploit the confidence of the interpretable channel's

predictions in building an alternative rule, that it is now no longer intelligible. Overall, `(B)LENs` is the only choice that allows the combination of both accuracy and an interpretable combination of the two channels, all while preserving the semantics of the channels as testified by the rule *number of red nodes* $\geq 1.23$ matching the expectations[7].

## C.5. More experiments on `AIDS`

In Table 3 we showed that a simple linear classifier on sum-aggregated node features can suffice for achieving the same, or better, performances than a plain SE-GNN. Even if the linear classifier is promoted for sparsity via weight decay regularization, the resulting model is still difficult to be interpreted, as it assigns non-negligible scores to a multiple of input features, making it difficult to extract a simple rule. For this reason, we extend the original set of node features by concatenating the value $1.0$ for each node. The role of this additional feature is to allow the linear classifier to easily access the graph-level feature *number of nodes*. We name the resulting updated dataset `AIDSC1`.

Then, we train the same models as Table 3, where we increase the sparsity regularization on both the SE-GNN and the linear model, to heavily promote sparsity. This is achieved by setting to $0.1$ the weight decay regularization for the SE-GNN, and to $0.01$ a $L_1$ sparsification to the linear classifier. The results are shown in Table 10, and show that under this strong regularization, SE-GNN struggles to solve the task. Conversely, the Dual-Channel GNN augmentation still solves the task, while providing intelligible predictions. In fact, by inspecting the weights of the linear classifier when picked by the Dual-Channel GNN, the resulting model weights reveal that the model is mainly relying on the count of nodes in the graph, as pointed out in Pluska et al. (2024). For completeness, we report in Table 12 the full weight matrix for `DC-SMGNN`.

Table 10: Dual-Channel GNNs solve `AIDSC1` even when prompted for strong sparsification, whereas plain SE-GNN achieves suboptimal predictions. Results are averaged only over seeds where the Dual-Channel GNN selects the linear classifier as the main channel. We indicate with superscript numbers the seeds left out from the analysis. The rule is extracted from the last column of Table 12 and averaged across nine seeds.

| Model | | AIDSC1 | | |
|---|---|---|---|---|
| | F1 | Channel | Rule | |
| `GIN` | $85 \pm 05$ | - | - | |
| `GISST` | $68 \pm 06$ | - | - | |
| `DC-GISST`[2] | $99 \pm 02$ | Rule | num nodes $\leq 12.66 \pm 0.18$ | |
| `GSAT` | $70 \pm 06$ | - | - | |
| `DC-GSAT`[4,10] | $99 \pm 02$ | Rule | num nodes $\leq 13.64 \pm 1.41$ | |
| `SMGNN` | $68 \pm 06$ | - | - | |
| `DC-SMGNN`[10] | $99 \pm 02$ | Rule | num nodes $\leq 13.89 \pm 3.07$ | |

## C.6. Dual-Channel GNN can improve the faithfulness of SE-GNNs

To measure the impact of the Dual-Channel GNN augmentation to plain SE-GNNs on the faithfulness of explanations, we compute Faith (Definition 5.1) for `SMGNN` and `GSAT` with their respective augmentations on `TopoFeature` and `GOODMotif`. Following Christiansen et al. (2023), we compute Faith for both the actual explanation and a randomized explanation obtained by randomly shuffling the explanation scores before feeding them to the classifier, and computing their ratio.

$$\text{Faith ratio} = \frac{\text{Faith}(\mathcal{E})}{\text{Faith}(R)} \tag{25}$$

where $R$ is the original explanation, and $\mathcal{E}$ a randomly shuffled explanation. The metric achieves a score of $1$ when the two values match, meaning the model is as faithful to the original explanation as a random explanation, whilst achieves a score of $0$ when the faithfulness of the original explanation is considerably higher than that of a random one.

We compute the metric over the entire validation splits, and extract hard explanations by applying a topK strategy as indicated in previous studies (Amara et al., 2022; Longa et al., 2024), where $k \in [0.3, 0.6, 0.9]$ for `GOODMotif`, and

---

[7]Since the number of nodes is discrete, any threshold value in $[1 + \epsilon, 2]$ corresponds to the rule $\geq 2$.

Table 11: Faithfulness of SE-GNNs and their augmentations for `TopoFeature` and `GOODMotif`. Following (Christiansen et al., 2023), we report the ratio between Faith computed over randomly shuffled explanations and original ones. Therefore, scores close to zero indicate better values.

| Model | TopoFeature
Faith ratio ($\downarrow$) | GOODMotif
Faith ratio ($\downarrow$) |
|---|---|---|
| GSAT | $0.42 \pm 0.10$ | $0.78 \pm 0.08$ |
| DC-GSAT | $0.39 \pm 0.43$ | $0.69 \pm 0.11$ |
| SMGNN | $0.65 \pm 0.28$ | $1.00 \pm 0.00$ |
| DC-SMGNN | $0.04 \pm 0.04$ | $1.00 \pm 0.00$ |

$k \in [0.05, 0.1, 0.2, 0.4, 0.8]$ for `TopoFeature`. Perturbations are limited to edge removals, and we force isolated nodes to be removed from the explanation. To compute Suf and Nec, we refer to the implementation of Azzolin et al. (2025) which requires a hyperparameter $b$ encoding the number of removals to apply at each perturbation. To obtain more robust results, we vary $b \in [0.01, 0.05, 0.1]$, corresponding to a number of perturbations equal to a $b$ percentage of the graph size. Then, the final Faith score is taken as the best Faith across $k$ and averaged across the values of $b$.

The final results are reported in Table 11 and highlight that for `TopoFeature`, where both DC-GNNs can exploit the interpretable channel (see Table 2), Dual-Channel GNNs can achieve considerable gains in faithfulness. `DC-GSAT`, in particular, achieves a better score but with a considerably higher standard deviation. By inspecting the raw scores, however, we see that across the values $b$, `DC-GSAT` scores $[1.00, 0.05, 0.12]$ whereas `GSAT` achieves $[0.50, 0.27, 0.48]$, indicating that $b = 0.01$ can be an unfortunate choice for this model as it may not bring enough perturbations to let the model change prediction. On the other values of $b$, however, the model achieves a significant gain in faithfulness ratio of almost an order of magnitude. On `GOODMotif`, instead, as the Dual-Channel GNNs does not have any advantage in using an interpretable model, we do not expect significant changes in the faithfulness scores as indicated in Table 11, where the only gain is due to higher variance.

We argue that the substantial gains in faithfulness mainly come from the ability of Dual-Channel GNNs to delegate each sub-task to the channel that can best handle it, i.e., learning the motif for the SE-GNN and the "$\geq$" rule to the linear classifier. In doing so, the underlying SE-GNN can better focus on highlighting the topological explanation, resulting in a more faithful explanation. This insight is supported by the analysis of the compactness of explanations provided in Appendix C.7, showing that indeed those SE-GNNs can better focus on the topological sub-task.

### C.7. Plotting Explanations

In this section, we aim to provide examples of explanations of SE-GNNs and DC-GNNs. For visualization purposes, we rely on an importance threshold to plot the *hard* explanatory subgraph over the entire graph. Such threshold is picked by plotting the histogram of importance scores and chosen in such a way as to separate regions of higher scores from regions of lower scores. We will analyze the following datasets:

**TopoFeature.** Fig. 5 and Fig. 6 present the histograms of explanation relevance scores for `GSAT` and `SMGNN` respectively. Overall, `SMGNN` achieves a better separation between higher and lower explanation scores, making it easier to select a proper threshold to plot explanations. Therefore, we will proceed to show explanation examples in Fig. 7 only for `SMGNN` for seed 1, picking as threshold the value $0.8$. Overall, the model succeeded in giving considerably higher relevance to edges in the motif, but failed in highlighting red nodes as relevant for predictions of class 1, hindering a full understanding of the model's decision process.

We proceed now to analyze the explanations extracted for the same samples for `DC-SMGNN`. First, we plot the histogram of explanation scores in Fig. 8, showing better sparsification than a plain `SMGNN`. For reference, we also plot the histogram for `DC-GSAT` scores in Fig. 9, where the same sparsification effect can be observed. Then, we report in Fig. 10 the explanations for the same graphs as in Fig. 7, showing that `DC-SMGNN` achieves better sparsification than a plain `SMGNN`. In fact, since the rule *at least two red nodes* is learned by the interpretable model, the underlying `SMGNN` now just looks at the topological motif, and indeed the substantially more sparse edge score reflects this behavior. Overall, `DC-SMGNN` explanations reflect more closely the actual predictive behavior of the underlying SE-GNN.

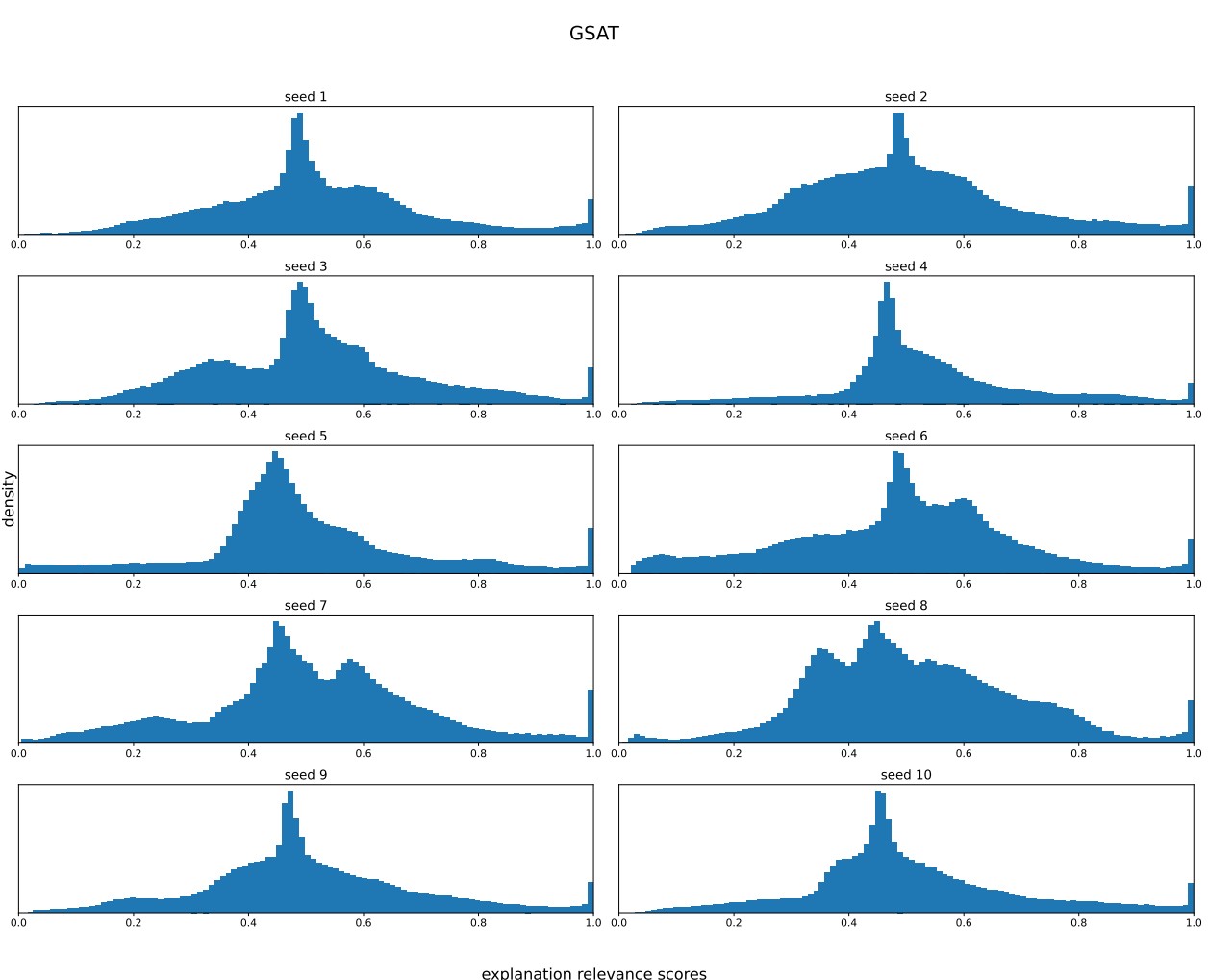

Figure 5: **Histograms of explanation relevance scores for `GSAT` on `TopoFeature`** (validation set). The model fails to reliably separate between relevant and non-relevant edges, making it difficult to select a proper relevance threshold.

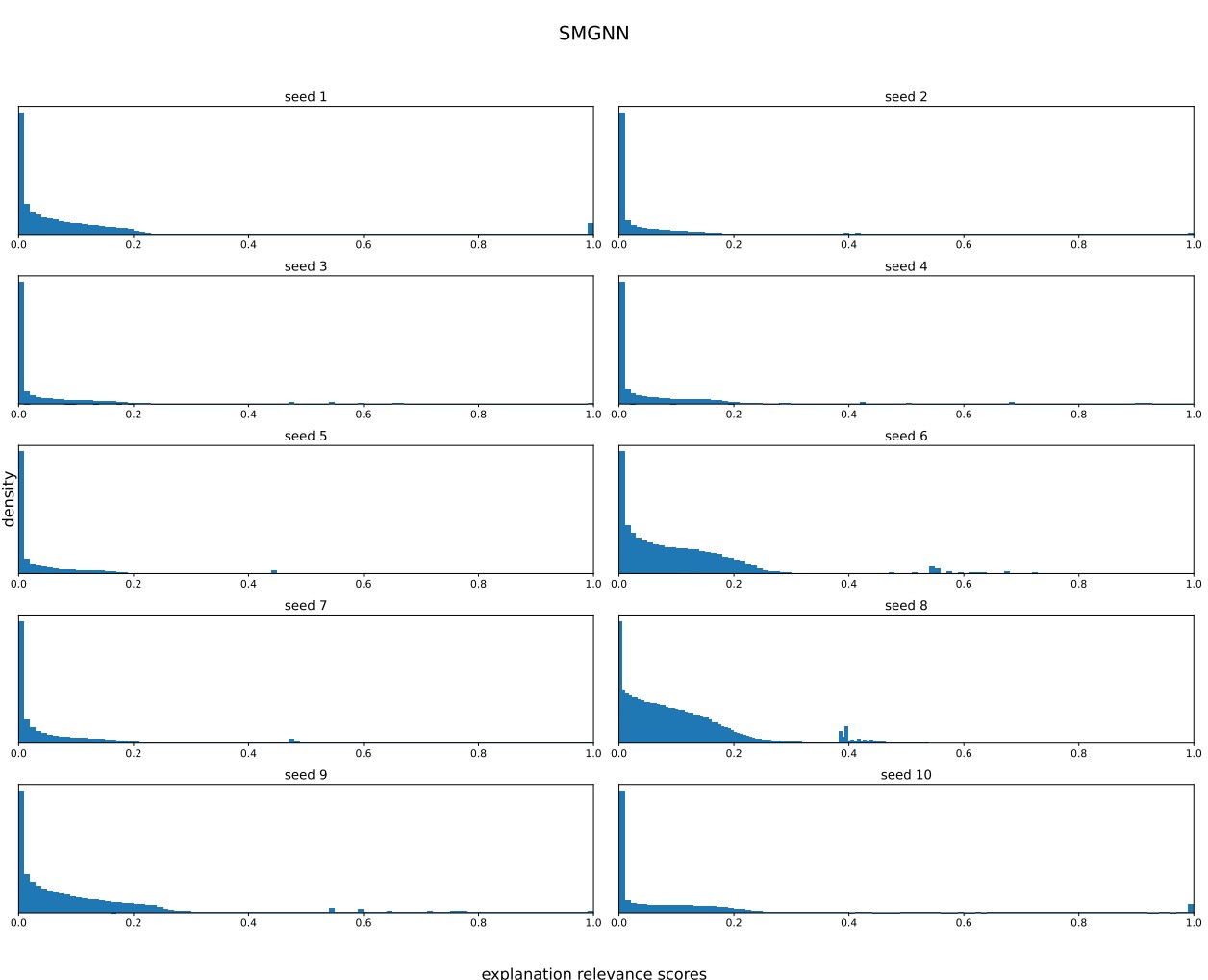

Figure 6: **Histograms of explanation relevance scores for SMGNN on TopoFeature** (validation set). The sparsification mechanism of SMGNN better separaMEs edges with higher importance than the rest of the graph.

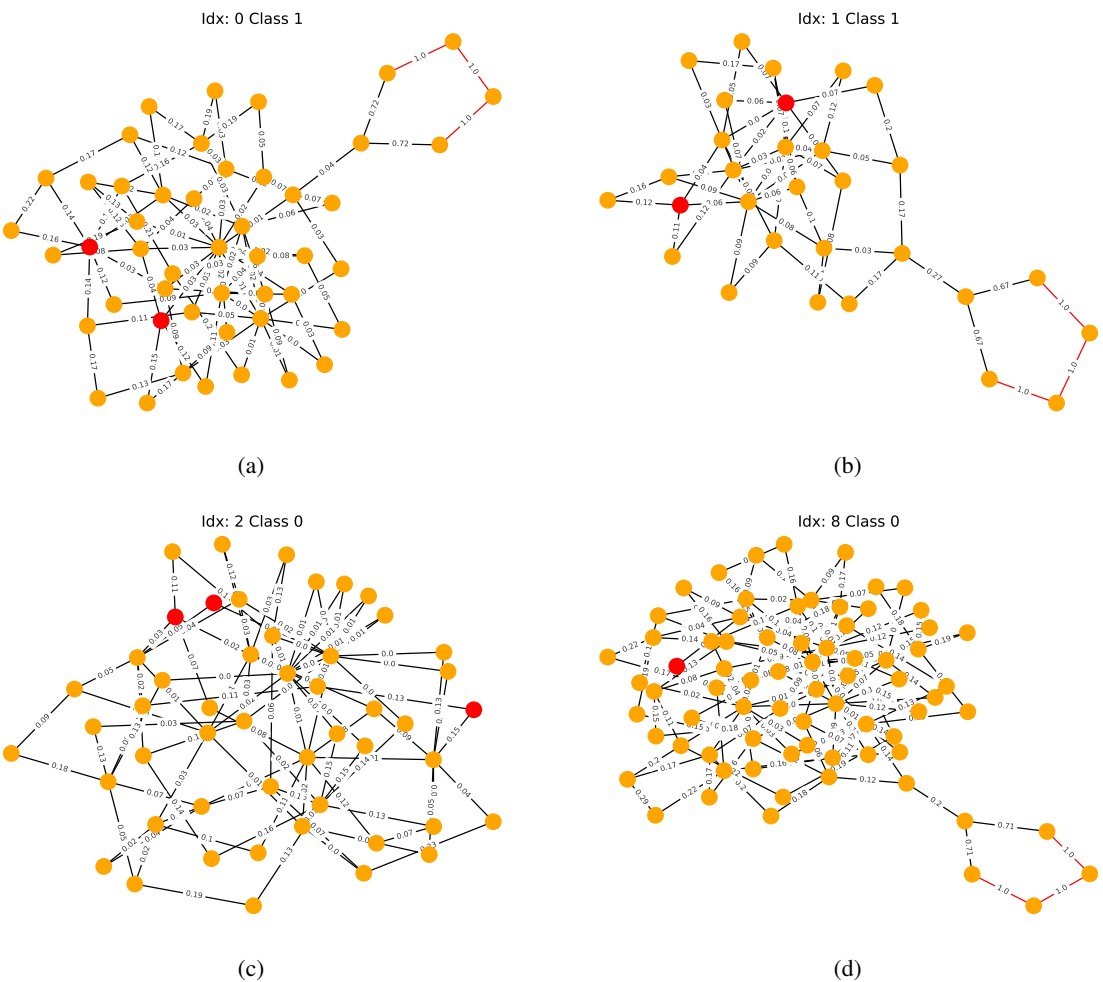

Figure 7: **Examples of explanations for SMGNN (seed 1) over TopoFeature**. Relevant edges are those with $p_{uv} \geq 0.8$ and are highlighted in red. Edges are annotated with their respective $p_{uv}$ score. Samples of class 1 must have both a cycle and at least 2 red nodes.

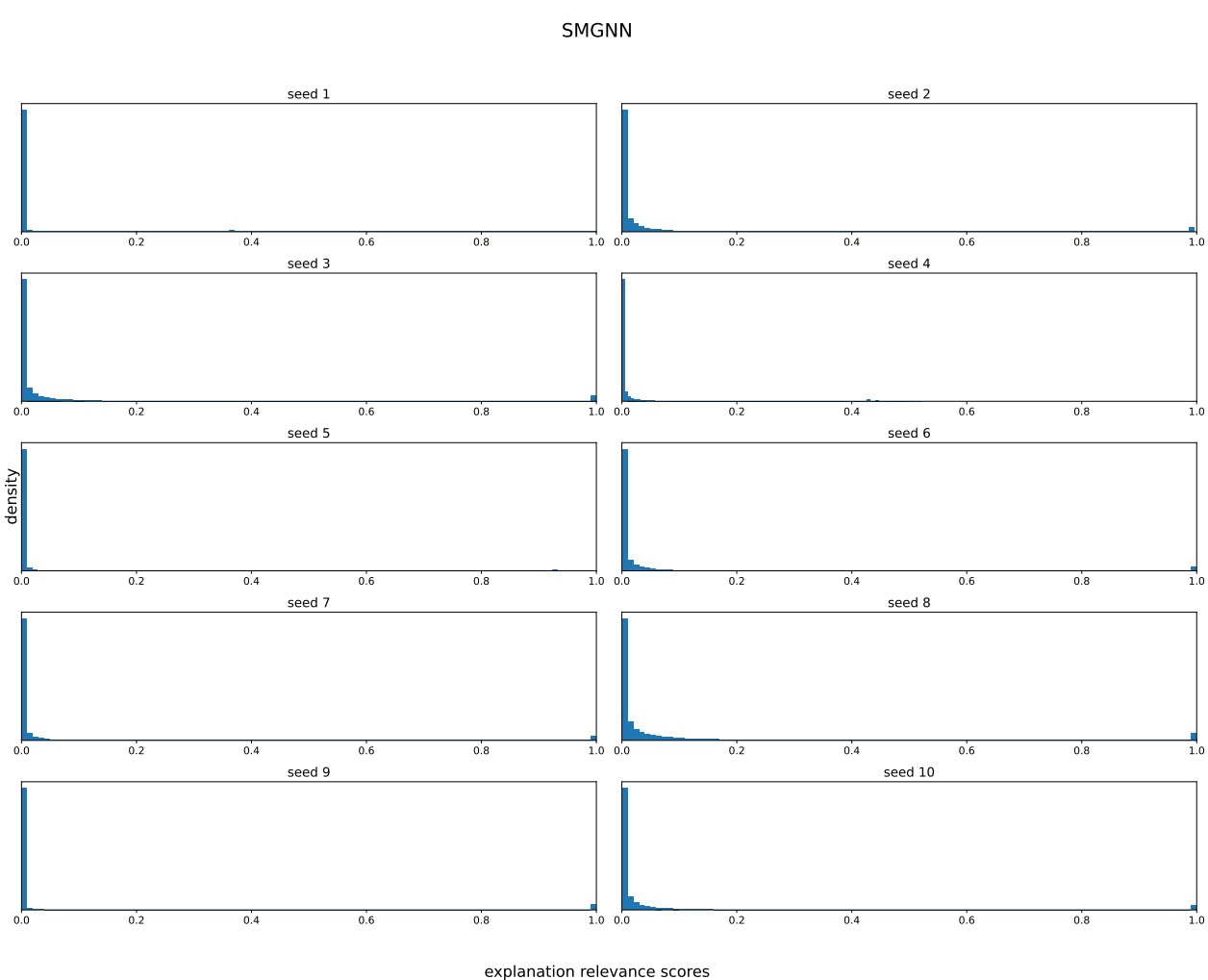

Figure 8: **Histograms of explanation relevance scores for `DC-SMGNN` on `TopoFeature`** (validation set). Since the underlying SMGNN is now only looking for the topological motif (as the rule *at least two red nodes* is learned by the interpretable model), the Dual-Channel GNN is allowed to sparsify all the other edges better, achieving more compact explanations. For seed 1, non-zero scores are clutter around 0.38.

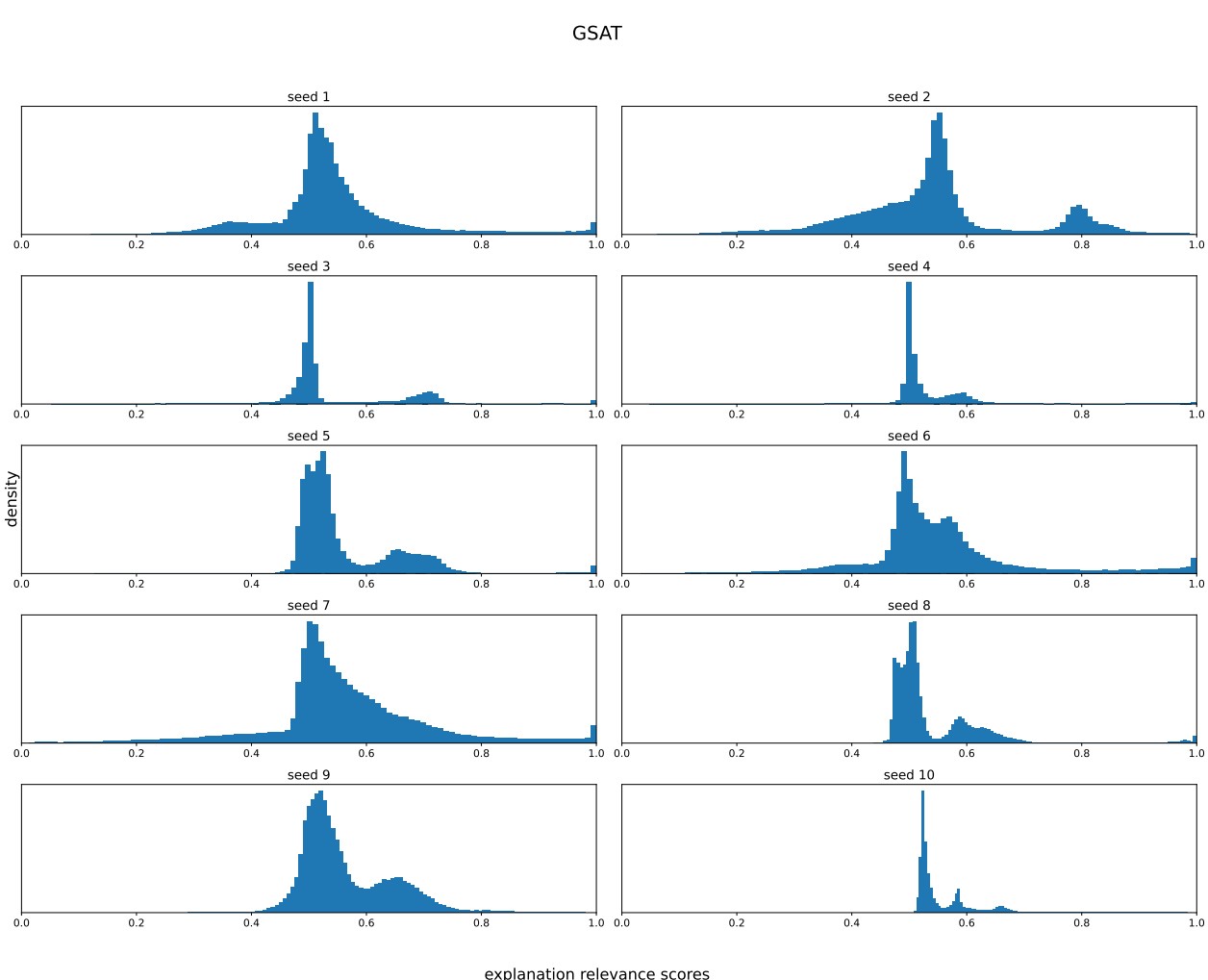

Figure 9: **Histograms of explanation relevance scores for `DC-GSAT` on `TopoFeature`** (validation set).

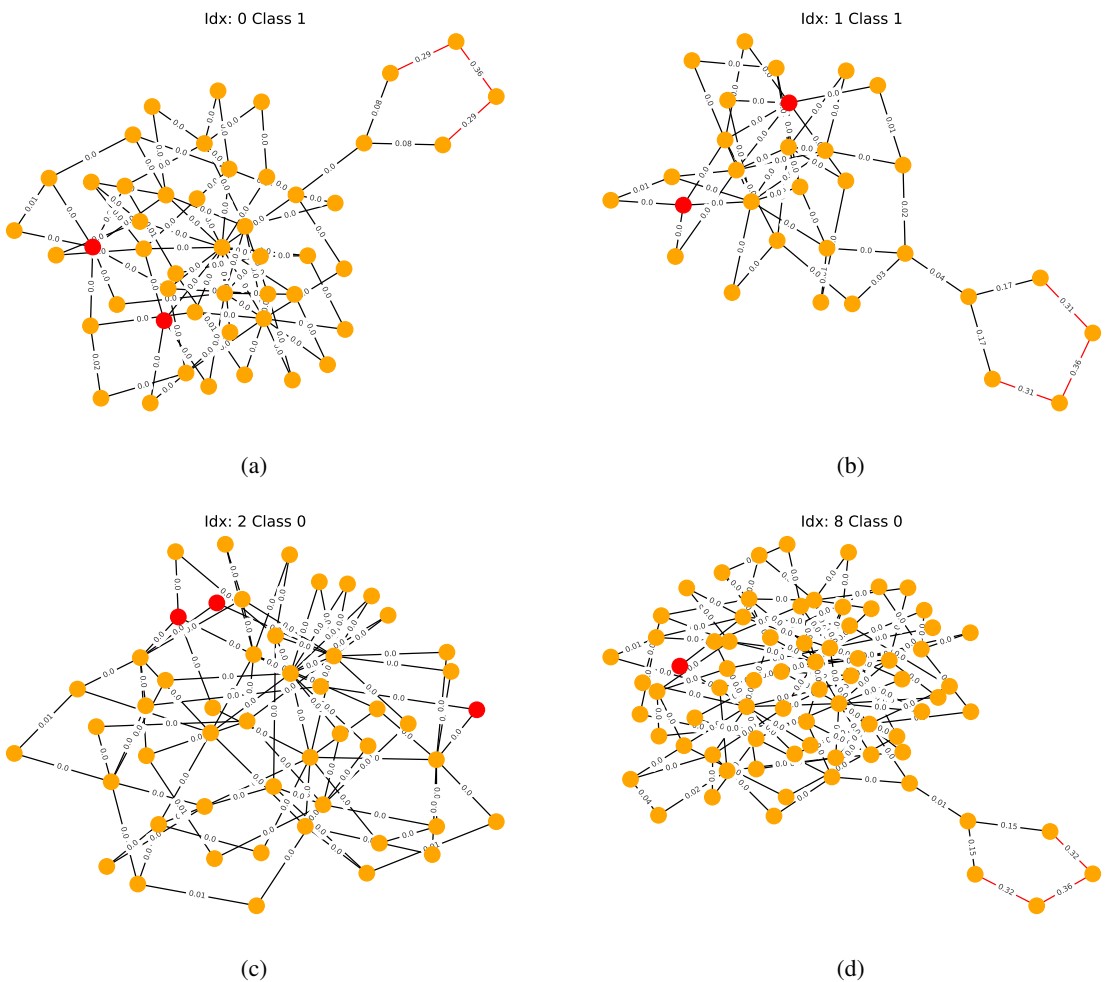

Figure 10: **Examples of explanations for `DC-SMGNN` (seed** 1**) over `TopoFeature`**. Relevant edges are those with $p_{uv} \geq 0.2$ and are highlighted in red. The threshold is picked by looking at the histogram in Fig. 8. Edges are annotated with their respective $p_{uv}$ score. Overall, DC-SMGNN achieves better sparsification than SMGNN (Fig. 7).

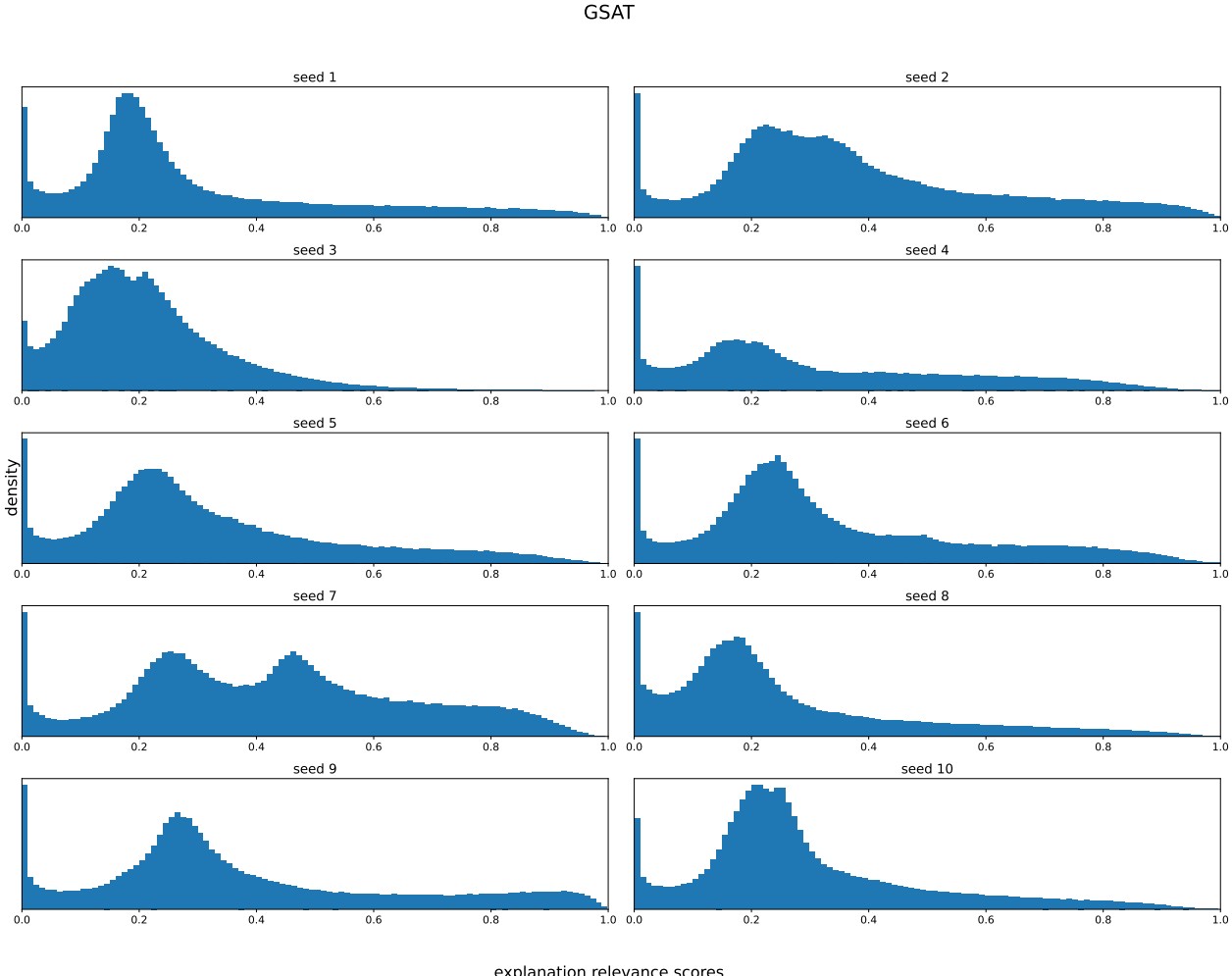

Figure 11: **Histograms of explanation relevance scores for `GSAT` on `RedBlueNodes`** (validation set). The model fails to reliably separate between relevant and non-relevant edges, making it difficult to select a proper relevance threshold. Specifically, most edges are assigned an importance close to $0.3$, which matches the uninformative prior $r$ selected during training.

**RedBlueNodes.** Fig. 11 and Fig. 12 show the edge relevance scores for GSAT and SMGNN respectively. Overall, the histograms show that both models fail to reliably identify a relevant subgraph with a consistently higher importance than irrelevant ones. Examples of explanations for both models, plotted in Fig. 13 and Fig. 14, confirm that subgraph-based explanations fail to convey actionable insights into what the model is predicting, as no clear pattern emerges from explanations. Conversely, both DC-GSAT and DC-SMGNN provide intelligible prediction by relying on a simple linear classifier encoding the ground truth rule *number of red nodes $\geq$ blue nodes* (see Table 2 and Fig. 3).

**AIDS.** Among each random seed, seed $8$ achieves a test F1 score of $1.0$, highlighting that the model is likely to have learned to just count the number of nodes in each graph, and to make the prediction based on such count (Pluska et al., 2024). This strategy is proven to be effective in this dataset, as highlighted in Appendix C.5 and Pluska et al. (2024). By plotting the histogram of explanatory scores in Fig. 15, and some examples of explanations in Fig. 16, we cannot unambiguously assess which rule the model is using for making predictions. Conversely, as shown in Appendix C.5, DC-GSAT can achieve the same performances while declaring that only node count statistics are being used for prediction.

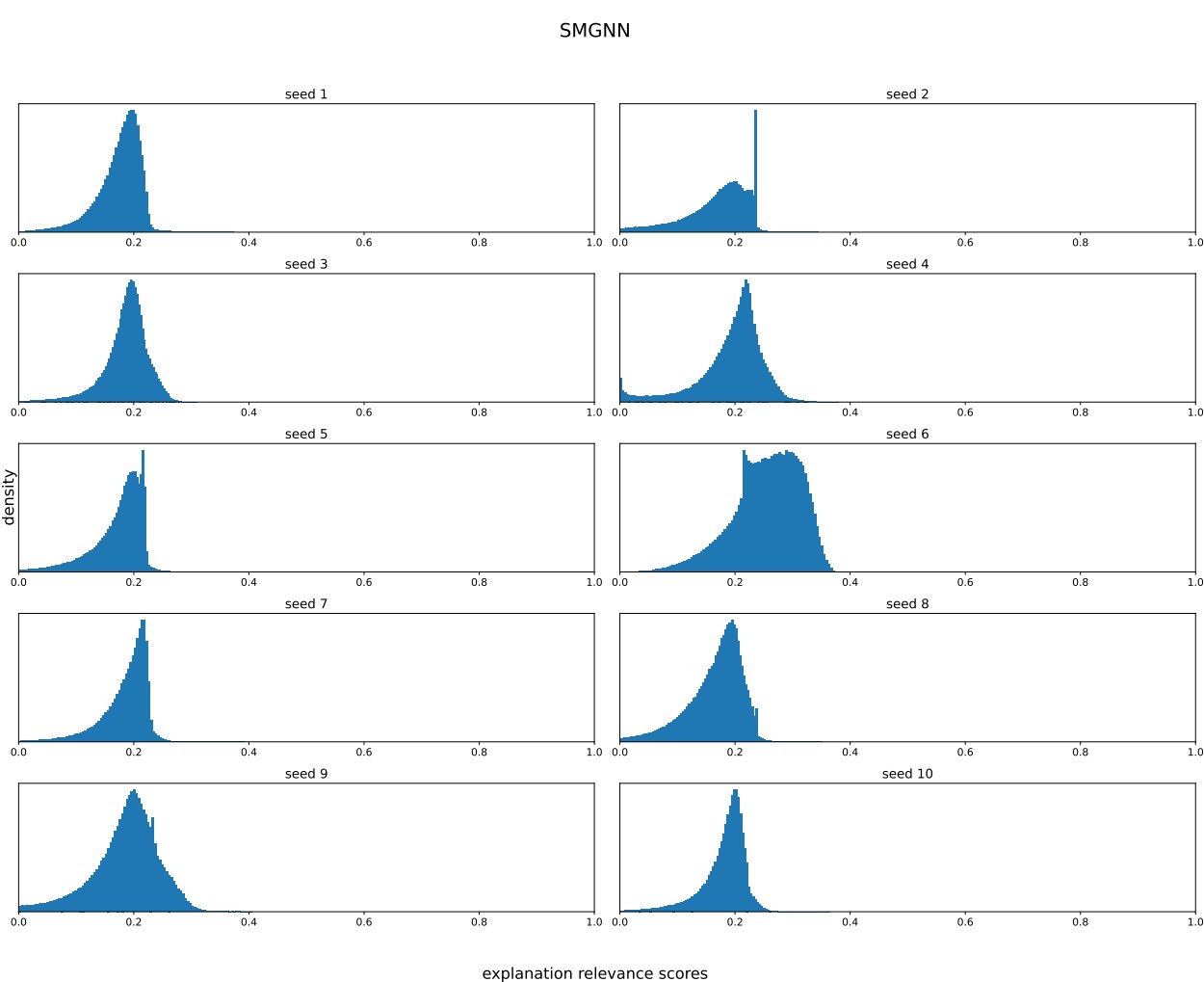

Figure 12: **Histograms of explanation relevance scores for `SMGNN` on `RedBlueNodes`** ((validation set)). The model assigns very cluttered scores to almost all edges, failing to highlight a subset that is reliably more relevant than the others, making it difficult to select an appropriate threshold for showing the explanations to consumers.

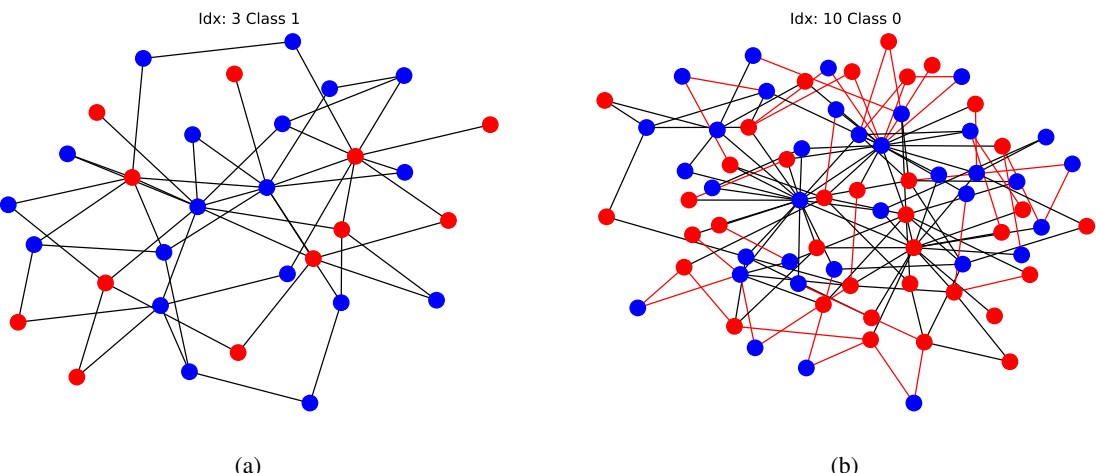

Figure 13: **Examples of explanations for `GSAT` (seed** 1**) over `RedBlueNodes`**. Relevant edges are those with $p_{uv} \geq 0.7$ and are highlighted in red. Edges are not annotated with their respective $p_{uv}$ score to avoid excessive clutter.

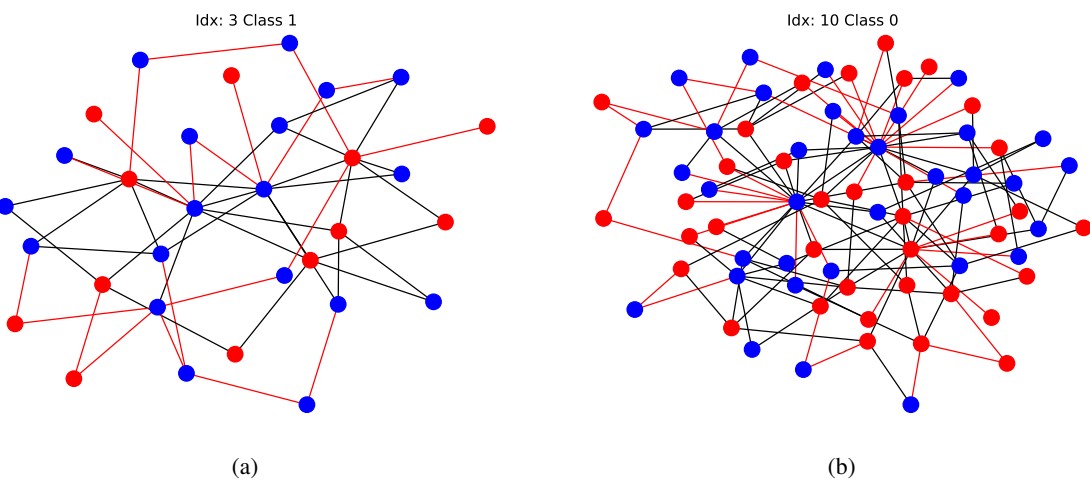

Figure 14: **Examples of explanations for `SMGNN` (seed** 1**) over `RedBlueNodes`**. Relevant edges are those with $p_{uv} \geq 0.2$ and are highlighted in red. Edges are not annotated with their respective $p_{uv}$ score to avoid excessive clutter.

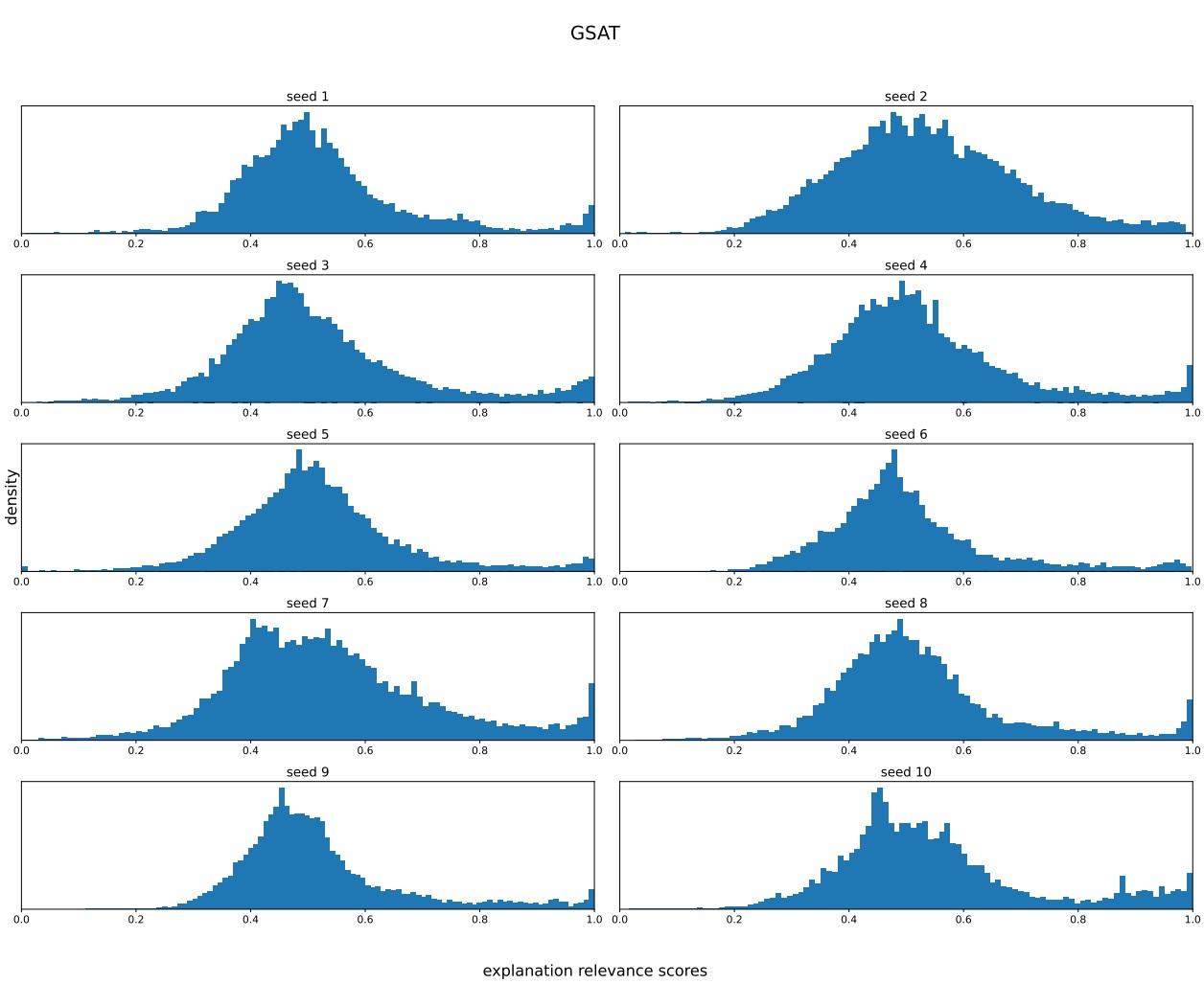

Figure 15: **Histograms of explanation relevance scores for GSAT on AIDS** (validation set).

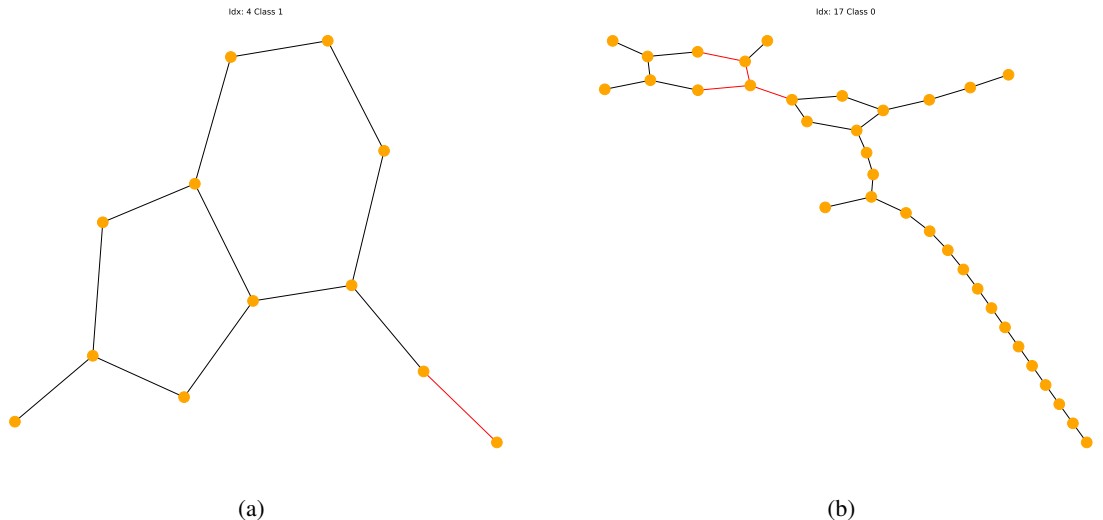

(a)                                                    (b)

Figure 16: **Examples of explanations for GSAT (seed** $8$**) over AIDS**. Relevant edges are those with $p_{uv} \geq 0.8$ and are highlighted in red. Edges are not annotated with their respective $p_{uv}$ score to avoid excessive clutter.

Table 12: Dual-Channel GNN's linear classifier weights for the `DC-SMGNN` experiment in Table 10. The $L_1$ sparsification effectively promotes the model to give considerably higher importance to the last input feature, corresponding to the number of nodes in the graph. Seed 10 is omitted as the linear classifier is not in use.

| Model | AIDSC1 | | | | | | |
|---|---|---|---|---|---|---|---|
| `DC-SMGNN`'s seed | | | Weights (W) | | | Bias (b) | - b / W[-1] |
| seed 1 | -6.1e-03 | -3.2e-02 | -9.5e-03 | -1.8e-02 | -3.8e-04 | | |
| | 7.8e-03 | -2.4e-04 | 7.0e-04 | 5.9e-04 | 1.4e-03 | | |
| | 3.9e-03 | 1.6e-02 | -4.4e-04 | 1.4e-03 | -6.9e-04 | | |
| | 4.1e-05 | -2.6e-02 | -7.0e-04 | 3.1e-04 | -9.6e-04 | 3.64 | 12.45 |
| | -3.8e-04 | 8.9e-04 | 1.1e-04 | 3.7e-05 | -2.2e-04 | | |
| | -3.3e-04 | -3.3e-04 | 9.7e-04 | 4.3e-04 | 1.4e-03 | | |
| | -8.6e-05 | 3.4e-04 | 4.1e-04 | 3.6e-04 | 1.1e-03 | | |
| | -1.7e-03 | 1.7e-03 | 3.4e-04 | -2.9e-01 | | | |
| seed 2 | -2.0e-02 | -3.0e-02 | 8.0e-03 | 7.3e-03 | -7.3e-04 | | |
| | 2.7e-02 | -6.5e-04 | 2.0e-02 | -4.9e-05 | -9.4e-04 | | |
| | 7.2e-02 | -4.0e-04 | -1.4e-04 | -2.6e-04 | -3.4e-04 | | |
| | 1.1e-03 | 3.0e-04 | 8.4e-06 | 1.3e-04 | -9.4e-04 | 3.63 | 12.57 |
| | -3.9e-05 | -9.3e-04 | -5.1e-04 | -1.4e-03 | 6.3e-04 | | |
| | 3.4e-04 | 1.3e-03 | 5.6e-04 | 1.2e-04 | -6.4e-04 | | |
| | 4.5e-05 | 3.8e-05 | 8.0e-05 | 1.6e-04 | -3.3e-04 | | |
| | 1.3e-04 | 7.1e-04 | 5.9e-04 | -2.9e-01 | | | |
| seed 3 | -1.2e-02 | -3.1e-02 | 3.2e-04 | 2.4e-04 | 3.6e-04 | | |
| | 2.4e-03 | -1.4e-04 | -4.4e-04 | 1.5e-06 | 1.8e-05 | | |
| | 6.1e-05 | 2.4e-04 | 6.5e-04 | -1.2e-04 | -2.3e-04 | | |
| | -6.4e-04 | -3.1e-04 | 1.0e-03 | -2.8e-03 | -1.2e-03 | 3.38 | 12.45 |
| | -3.9e-05 | 1.1e-03 | -5.2e-04 | -6.0e-04 | -1.1e-03 | | |
| | -4.1e-05 | 2.0e-04 | -3.6e-04 | -9.2e-04 | -7.6e-04 | | |
| | -3.4e-05 | -1.3e-04 | -8.7e-04 | -8.2e-04 | -1.4e-03 | | |
| | 5.4e-04 | 6.6e-04 | -3.3e-04 | -2.7e-01 | | | |
| seed 4 | -9.7e-03 | -1.6e-02 | -7.9e-03 | -2.0e-02 | 8.0e-04 | | |
| | 4.8e-03 | -5.9e-04 | 6.0e-05 | 3.9e-04 | -1.1e-03 | | |
| | -3.6e-04 | 6.3e-04 | -1.7e-03 | 3.2e-04 | -1.0e-03 | | |
| | 4.4e-04 | -3.4e-02 | 7.9e-04 | -8.5e-04 | -6.0e-04 | 3.42 | 12.04 |
| | 4.7e-04 | -2.8e-04 | -9.4e-04 | -5.9e-04 | -1.0e-03 | | |
| | 1.2e-05 | -7.1e-05 | -5.1e-05 | 9.3e-04 | 1.3e-03 | | |
| | 1.5e-04 | -1.3e-04 | -4.9e-04 | 5.1e-04 | 7.9e-04 | | |
| | 2.5e-05 | -4.0e-04 | -4.4e-04 | -2.8e-01 | | | |
| seed 5 | -1.7e-01 | -2.0e-01 | -1.3e-01 | 1.2e-04 | -8.6e-04 | | |
| | -1.2e-03 | 2.1e-04 | -2.0e-04 | -1.4e-03 | -5.7e-04 | | |
| | 1.3e-04 | -5.2e-04 | -3.5e-04 | -9.1e-04 | 4.6e-04 | | |
| | -6.7e-04 | 3.6e-04 | 1.0e-03 | -6.8e-04 | -4.0e-04 | 4.48 | 22.18 |
| | -4.7e-04 | -9.2e-04 | 8.1e-04 | -5.2e-04 | -1.9e-06 | | |
| | -2.4e-04 | -5.2e-04 | 2.0e-04 | 4.5e-04 | -1.3e-03 | | |
| | 5.5e-04 | 2.5e-04 | -5.5e-04 | 6.2e-04 | -1.0e-03 | | |
| | 4.2e-04 | 2.1e-04 | 8.9e-04 | -2.0e-01 | | | |
| seed 6 | -1.0e-02 | -6.9e-02 | -2.0e-03 | -1.3e-02 | -4.7e-03 | | |
| | 1.5e-02 | -1.1e-03 | 5.1e-04 | 8.4e-04 | 7.4e-04 | | |
| | -8.2e-04 | -3.4e-04 | -1.3e-03 | 6.7e-04 | -8.1e-07 | | |
| | 1.3e-03 | -1.7e-02 | 6.9e-04 | 2.7e-04 | 8.7e-04 | 3.90 | 15.07 |
| | -8.5e-05 | 3.1e-04 | -3.1e-04 | -5.3e-04 | -9.0e-04 | | |
| | -4.1e-04 | 1.2e-03 | 5.8e-04 | -4.2e-04 | 1.6e-03 | | |
| | -5.1e-04 | 9.4e-04 | 4.4e-04 | 8.4e-04 | 6.4e-05 | | |
| | 2.4e-04 | -1.7e-04 | -9.2e-04 | -2.6e-01 | | | |
| seed 7 | -1.8e-02 | -2.7e-02 | -5.5e-03 | -1.6e-02 | 4.9e-04 | | |
| | -4.3e-03 | -8.5e-04 | 1.6e-04 | -8.3e-04 | 9.1e-04 | | |
| | -5.5e-04 | -1.1e-03 | -6.8e-04 | -1.6e-04 | -1.4e-03 | | |
| | 9.0e-04 | -3.5e-02 | 2.6e-04 | 1.3e-03 | 3.0e-04 | 3.63 | 12.32 |
| | -5.3e-05 | -1.0e-03 | -1.0e-03 | -4.5e-04 | 7.9e-04 | | |
| | 6.3e-04 | 1.3e-03 | -5.1e-04 | 8.9e-04 | -4.7e-04 | | |
| | 2.0e-03 | 1.0e-03 | -3.9e-04 | -1.0e-03 | -1.0e-04 | | |
| | 4.6e-04 | 1.0e-03 | -6.5e-04 | -2.9e-01 | | | |
| seed 8 | -1.0e-02 | -2.2e-02 | 2.0e-03 | -1.6e-02 | 3.1e-04 | | |
| | 5.0e-04 | -1.2e-03 | -3.9e-04 | 7.8e-04 | -5.8e-04 | | |
| | 9.9e-05 | -8.4e-04 | 7.6e-04 | -3.9e-04 | 2.0e-04 | | |
| | 7.2e-04 | -2.3e-02 | -5.3e-04 | -4.5e-04 | 7.0e-04 | 3.47 | 12.18 |
| | 5.9e-04 | -1.1e-04 | 6.5e-04 | -6.3e-04 | 7.9e-06 | | |
| | -5.8e-04 | -3.3e-04 | -5.5e-04 | -1.2e-03 | -9.8e-05 | | |
| | -7.0e-05 | 3.1e-04 | -3.4e-04 | -2.0e-03 | 7.8e-05 | | |
| | -3.5e-04 | -8.5e-04 | 3.3e-05 | -2.9e-01 | | | |
| seed 9 | -1.4e-02 | -3.1e-02 | 5.7e-03 | -2.1e-02 | -2.2e-04 | | |
| | 8.5e-03 | 3.0e-05 | 1.3e-04 | -6.1e-04 | 5.7e-04 | | |
| | 1.2e-04 | -2.5e-03 | -1.3e-03 | -1.8e-04 | 1.7e-03 | | |
| | 2.5e-04 | -2.7e-02 | 6.9e-04 | 2.5e-04 | 1.1e-03 | 3.59 | 13.73 |
| | -5.1e-04 | -9.9e-04 | 1.5e-04 | 1.1e-04 | 2.4e-04 | | |
| | -9.5e-04 | -8.2e-04 | 1.6e-04 | -3.8e-04 | -6.3e-04 | | |
| | -5.5e-04 | 1.7e-03 | 1.0e-03 | -1.1e-04 | 6.4e-04 | | |
| | 7.1e-04 | -1.9e-04 | -4.5e-04 | -2.6e-01 | | | |

