# OpenReview forum: "Beyond Topological Self-Explainable GNNs: A Formal Explainability Perspective"
_ICML.cc/2025/Conference — ICML 2025 poster_

### Official Review · Reviewer_rjdA · 2025-03-05

**Overall Recommendation:** 4

**Summary:**

In this paper, the authors argue that existing SE-GNNs generate Trivial Explanations and advocate for designing GNN explanation methods capable of producing Prime Implicant and faithful explanations. After conducting theoretical analysis, they propose DC-GNNs, a dual-channel model that incorporates a non-topological rule extractor. They evaluate their method on several datasets and conclude that DC-GNNs perform on par with or better than standard SE-GNNs.

## update after rebuttal

The rebuttal clarified key concepts I had misunderstood in my initial review. Therefore, I have raised my score from weak reject (2) to accept (4).

**Claims And Evidence:**

Theorem 3.2 suggests that existing SE-GNNs provide only Trivial Explanations. While this is theoretically correct, it does not always hold in practice: (1) As shown in Figure 3 of GSAT, existing SE-GNNs identify all important edges rather than strictly minimal explanations. (2) In multi-loss optimization, the hyperparameter $\lambda$ used to constrain $R$ is typically small (to maintain stable training and ensure predictive accuracy). Consequently, the model may retain more edges to preserve predictive performance, even at the cost of some conciseness (cf. Figure 10 of GSAT). Considering that the critique of existing SE-GNNs throughout this work is primarily based on Theorem 3.2, a significant discrepancy between theory and practice would undermine the impact of this paper.

**Essential References Not Discussed:**

I think the paper is original as I have not seen anything like it before.

**Experimental Designs Or Analyses:**

- Limited metrics: This paper focuses on GNN explanation, yet the primary evaluation metrics only assess predictive accuracy. A more comprehensive comparison between DC-GNNs and SE-GNNs appears only in Appendix C.3, and even there, it is limited to just two synthetic datasets.

- Unfair experimental settings: The authors introduce two new datasets, RedBlueNodes and TopoFeature, where labels are constructed using additional color information. However, the design of DC-GNNs inherently biases the model towards discovering color information, leading to an unfair comparison.

- Inappropriate parameter choices: According to the GSAT authors’ recommendations, the parameter r should be set to $r \geq 0.5$. However, the authors set  $r=0.3$ for RedBlueNodes (line 910), deviating from the guideline. Additionally, in GSAT, the hidden size of GIN for the Graph-SST2 dataset is 64, whereas the authors set it to 300, which may introduce inconsistencies in comparison.

- Limited baselines: The baseline selection is quite limited, as the authors include only two baselines, one of which is a 2020 ArXiv paper with just 15 citations. Given that the paper extensively discusses sufficiency and necessity, it would be more appropriate to include causal-based baselines for comparison. For instance, Wu et al. (2020) is cited (line 300), so it would be reasonable to compare against it. Furthermore, DC-GNNs incorporate node features, yet multiple prior works also consider this aspect, such as GNNExplainer [1] and CAL [2]. So, comparing DC-GNNs with these methods might be a good way to verify the effectiveness of DC-GNNs.

- Misleading experimental analysis: On real-world datasets, the advantage of DC-GNNs is limited and can sometimes even lead to a decline in performance. However, the authors claim that DC-GNNs perform on par with or better than plain SE-GNNs. Additionally, in Table 4, the authors state that DC-GNNs generalize better to OOD than plain SE-GNNs. However, on Motif OOD_2, DC-GNNs exhibit significantly worse performance compared to plain SE-GNNs.

[1] GNNExplainer: Generating Explanations for Graph Neural Networks, NeurIPS 2019

[2] Causal Attention for Interpretable and Generalizable Graph Classification, KDD 2022

**Methods And Evaluation Criteria:**

- Gap between method and theoretical analysis: From my perspective, the proposed method does not appear to have a strong connection to its theoretical analysis. While building on existing work, DC-GNNs incorporate a non-topological perspective by analyzing node features. In other words, DC-GNNs perform better on certain types of datasets, but there is no evidence that they can generate PI explanations.
- Gap between benchmark datasets and theoretical analysis: The dataset used by the authors seems unrelated to the theoretical analysis. None of the datasets have a strong connection to FOL and Figure 1, creating a significant gap between the theoretical analysis and the experimental evaluation.

**Other Comments Or Suggestions:**

- I suggest that the authors provide more details about Figure 1, as it is crucial for readers to fully understand the paper. I would appreciate it if the authors could clarify the meanings of the solid and dashed lines in Figure 1, illustrate what the raw graphs $G$ looks like, and specify whether the two cases presented in the figure correspond to positive or negative samples.
- I see that the authors use faithfulness to evaluate the effectiveness of self-interpretable GNNs (e.g., GSAT). However, I think this is inappropriate because, according to Eqs. 7 and 8, Suf(R) and Nec(R) are computed based on $\Delta(G, G^{\prime})$ . These metrics are suitable for post-hoc GNN explanation methods, which are trained to align with the predictive results of the original graph $G$. However, self-interpretable GNNs are trained to make accurate predictions when $R$ is provided as input and do not need to align with the predictions of $G$. As a result, using Eqs. 7 and 8 to assess the effectiveness of self-interpretable GNNs suffers from a distribution shift problem. That said, I remain open-minded on this issue, as I am aware that none of the recent works have identified this problem.

**Other Strengths And Weaknesses:**

I appreciate the authors’ effort in writing such a comprehensive paper -- it is the longest paper in my batch and the one I spent the most time reviewing. I am worried that I misunderstood some parts. If the authors can address my concerns during the rebuttal phase, I would be open to reconsidering and potentially increasing my score.

**Questions For Authors:**

- In Figure 2, why do SE-GNNs fail to capture non-topological patterns? Some SE-GNNs use node features to determine edge weights, which suggests that they might have the potential to discover non-topological patterns.
- If the white-box rule extractor (line 30) is a sparse linear model, why can it be described as white-box?
- What does ± mean in all tables? Is the number after ± at the decimal level or the integer level?
- Which metrics did you use in Table 4?

**Relation To Broader Scientific Literature:**

Investigating GNN explanations is crucial for expanding the applicability of GNNs in high-stakes scenarios.

**Theoretical Claims:**

Based on the provided proof sketch, Theorem 3.2 is correct. However, starting from Theorem 3.4, I can't understand why a single $e \in E$ can be a Trivial Explanation for $g^{\prime}(G)$, which prevents me from checking the correctness of the subsequent proofs.

---

> ### Author Rebuttal · Authors · 2025-03-31
>
> Thank you for the feedback. See our comments below.
>
> **Summary clarification:** We are *not* advocating for extracting PI explanations from SE-GNNs. As per the Abstract and Sec 6 (line 244), PIs can be intractable to compute and as large as the input. Our analysis is not meant as criticism. In fact, we advocate for enhancing SE-GNNs rather than replacing them.
>
> We renamed “Trivial Explanation” to “Minimum Explanations”, which sounds less negative.
>
> **Theory-practice gap:** Thm 3.2 does not guarantee that SE-GNNs always achieve TEs in practice, but it shows TEs are necessary and sufficient for minimal true risk *assuming perfect predictive accuracy* for any $\lambda>0$. Optimization may not reach this solution, as also discussed in 666-673 Appx A.1. Still, the result remains impactful: even if Thm 3.2’s conditions are not met, SE-GNNs in Table 1 optimize for small (ideally the smallest) label-preserving subgraphs and our theoretical analysis applies identically, as such explanations are still neither PI nor faithful in general. Figure 3 of GSAT supports the analysis above: TEs for that task would comprise 3 houses, and GSAT indeed highlights 3 houses with only a few additional edges.
>
> **Method-theory gap:** Our analysis provides a formal argument for using (and enhancing) subgraph-based explanations, and it directly informs the design of DC-GNNs.
>
> **DC-GNNs & PI:** DC-GNNs are not designed to output PIs but to leverage SE-GNN strengths in existential tasks, combined with an interpretable model for non-existential properties
>
> **Datasets not related to FOL:** Synthetic datasets are tightly linked to FOL. TopoFeature matches the formula $\exists a,b,c,d. Cycle(a,b,c,d) \land \exists^{\ge 2}x.Red(x)$. Motif follows a similar structure, while RedBlueNodes is an instance of FOL to counting quantifiers.
>
> Figure 1 is an illustration to provide intuition.
>
> **Why a single edge is a TE for $g′$?** $g′(G)$ checks for the existence of an edge for each node. Thus, a subgraph consisting of only one edge (seen alone) indeed satisfies this constraint. We will clarify.
>
> **Mostly assess accuracy:** We also evaluate DC-GNNs by measuring faithfulness and rule quality by testing how well they generalize OOD (Table 4), their dependability (Table 2), and their conciseness (C.4). Plausibility is unsuitable, as the ground-truth explanation is either unknown or not a pure subgraph, a problem for all models beyond subgraph explanations (Bechler-Speicher et al. 2024; Pluska et al. 2024).
>
> **Datasets are biased for colors:** The synthetic datasets are explicitly designed to test DC-GNN's ability to decompose the task between the two channels. To showcase its generality, we also test it on standard benchmarks where color information is not present.
>
> **Hyperparams choice:** We verified that using $r \in \\{0.5,0.7\\}$ for GSAT doesn’t change our results, which will be added in the revision. GIN performs better with a hidden size of 300, aligning with Gui et al. 2022. Keeping this value fixed ensures fair comparison across our experiments.
>
> **Limited baselines:** We added Wu et al. (2020) as a new baseline. Please refer to the answer to 3Jwx for the results.
>
> **Prior works also consider node features:** Including node feature relevance in topological explanations is not the same as extending them with rules:
>
> *Example*: for a model trained on RedBlueNodes, GNNExplainer and CAL will highlight individual nodes' color features. DC-GNNs, instead, will return the formula *Red $\ge$ Blue*, which is far more compact and intelligible.
>
> **Misleading experiments:** In real-world tasks, performance drops only 1% AUC for BBBP, which is within std dev. Elsewhere, DC-GNNs match or exceed baselines.
>
> For OOD2 in Motif, the drop likely stems from OOD performance instability in models without OOD regularization (Section 7, A1) and not the unused side channel (Table 2), as shown by the higher std dev.
>
> **Fig. 1:** It shows a positive instance G composed of a 3-clique for both formulas. Solid (resp. dashed) edges belong (resp. do not belong) to the explanation depicted in each column. We will clarify the caption.
>
> **Faithfulness is inappropriate:** We follow previous work computing SUF and NEC by feeding G and G’ into the full model, including the explanation extractor. Thus, the classifier evaluates $R'=q(G’)$, not $G’$, alleviating the OOD issue. Whether those metrics are suboptimal for SE-GNNs is beyond the scope of our contribution.
>
> **Q1:** SE-GNNs are bound to explain non-topological patterns by subgraphs, which can be ambiguous.. E.g., in Figure 2 the SE-GNN highlights the red nodes by marking their incident edges, and it’s unclear if what matters are the edges, the nodes, their multiplicity, etc.
>
> **Q2:** One can understand which (few) features are most responsible for a prediction and extract rules from them (see Appx B.3) without resorting to post-hoc methods.
>
> **Q3:** It’s the integer value of standard dev.
>
> **Q4:** Accuracy. We will clarify this.

---

> > ### Comment · Reviewer_rjdA · 2025-04-02
> >
> > Thank you for your thoughtful response and for clarifying the points I previously raised. Now I have a better understanding of the paper and will increase my score :)
> >
> > Side note: While I feel that the final practical takeaway -- explicitly augmenting SE-GNNs with alternative modalities of explanations -- is somewhat trivial, the analysis between TE and PI contributes meaningful insights to the GNN explanation community.

---

### Official Review · Reviewer_3Jwx · 2025-03-11

**Overall Recommendation:** 4

**Summary:**

This paper investigates the properties and limitations of explanations generated by Self-Explainable Graph Neural Networks (SE-GNNs). The authors formalize SE-GNN explanations as Trivial Explanations (TEs) and compare them to Prime Implicant (PI) and faithful explanations. They find that TEs, while effective for motif-based tasks, can be less informative and misaligned with a common definition of faithfulness in general graph classification. The paper highlights that TEs match PI explanations in motif-based prediction tasks when the presence or absence of one particular motif has a causal relationship with the class label. The authors proceed to propose Dual-Channel GNNs (DC-GNNs), which combine SE-GNNs with non-relational predictors, to improve explanation compactness and performance. DC-GNNs adaptively employ both channels, allowing the SE-GNN to focus on topological motifs. Experiments on synthetic and real-world datasets show that DC-GNNs perform competitively with SE-GNNs, often with more succinct rules. The study reveals that TEs can be ambiguous and unfaithful, limiting their utility in certain applications. Overall, the research formally analyzes SE-GNN explanations and introduces a novel architecture to address their shortcomings. The findings suggest that integrating non-topological information enhances the quality and interpretability of GNN explanations.

**Claims And Evidence:**

The paper makes the following claims:

(1) Formal characterization of the class of explanations SEGNNs optimize for, namely "trivial explanations" (TEs).
Evidence: the evidence is not convincing, and this is an overclaim. The above claim only holds for the loss functions given in Table 1. Several SE-GNNs do not use these losses. Or they use these losses but with more sophisticated subgraph extraction methods.

(2) Formal comparison of TEs, PIs, and faithful explanations.
Evidence: this is done well and relates these different classes of explanations to each other. In my opinion, this is the main contribution of the paper.

(3) DC-GNNs are proposed as an alternative to existing SE-GNNs. They are claimed to be better empirically.
Evidence: The evidence here is empirical and weak, also because specific benchmarks are not used in the paper.

**Essential References Not Discussed:**

None.

**Experimental Designs Or Analyses:**

The experimental design seems sound. However, comparing the models using common topological benchmarks such as BA-2MOTIFS and MUTAG_0 would be better. A worry about the proposed DC-GNN model is that it will overfit or latch on to node features (through the second channel) and, therefore, have more difficulty finding the topological features.

**Methods And Evaluation Criteria:**

The paper proposes a new class of models (DC-GNN) compared empirically to some existing SE-GNNs. The authors also investigate some properties of the new class, such as how well the model can indicate which type of features it uses (topological or feature-based).

**Other Comments Or Suggestions:**

Please rethink the term "trivial explanation." As you wrote in the paper, they are often preferable, and indeed, your proposed method also attempts to extract them. They can be very complex. Indeed, in some cases finding a specific topological motif is an NP-hard problem.

You also contrast TEs with the "measures" of faithfulness which have their own problems.

So, the authors might want to show more critical distance to these (rather arbitrary) notions of faithfulness and what a good explanation is when writing the paper.

Please also consider, as written before, running experiments on existing benchmarks and using methods that do not follow the loss functions given in Table 1.

**Other Strengths And Weaknesses:**

The paper consists of three parts, each with its own contribution.

First, the connection between SE-GNNs and TEs. This is an interesting result but it should be clarified that Theorem 3.2 does not generally hold for all SE-GNNs. Even in the statement of the theorem, this can be easily overlooked. Other than the imprecise claims made in the paper‘s language, the theoretical results are  a strength of the paper.

I have a bit of a problem with the term "trivial explanation," as it might lead people to associate these explanations with "useless" or "uninteresting" explanations. This is, of course, not the case. In general, the paper has a negative tone about what they call TEs throughout the first part of the paper, showing that TEs have undesirable properties (from their point of view) compared to PIs and faithfulness measures. However, before introducing their own method, they completely change course, saying that TEs are often the preferable topological explanations (because faithful and PIs might grow very large and unhelpful for many graph problems) and then introducing a model that augments TEs with feature-based explanations.  Therefore, the first (theoretical) and the second (empirical) parts are somewhat disconnected.

The empirical evaluation is sound but misses several benchmark datasets, such as BA-2MOTIFS. It is strange since their proposed method is a SE-GNN with an additional feature-based explanation channel. It would also be good to compare to existing methods such as https://link.springer.com/article/10.1007/s10994-024-06576-1. Here, I assume that the method (which also incorporates features) would extract the topological motif and connect nodes whose features are important via edges to this topology. The method is also not one that falls into Table 1.

**Questions For Authors:**

- Why did you not use datasets such as BA-2MOTIFS?
- How is your method better than a SE-GNN that extracts a subgraph, and on top of which, you can run a feature attribution method?
- How does your theoretical analysis (part 1 of the paper) fit to your proposed method? Why are these two parts in the same paper?

**Relation To Broader Scientific Literature:**

A substantial contribution of the paper is the theoretical comparison of TEs, PIs, and faithfulness measures. This has not been done before and brings interesting insights into the otherwise mostly empirically driven research direction of explainable GNNs. The paper is therefore of interest to a XAI for GNN  subcommunity and well integrated into the literature.

**Theoretical Claims:**

The paper provides theoretical results—one theorem connecting SE-GNNs with specific loss functions to TEs.
Moreover, several theorems compare TEs, PIs, and faithfulness measures. The theoretical results are correct. However, some parts of the paper overclaim the implications of some of these theorems. The theory does not support the claim that all SE-GNNs are restricted to TEs.

---

> ### Author Rebuttal · Authors · 2025-03-31
>
> We thank the reviewer for their feedback. We are glad they found our analysis of interest to the community. Below, we address their remarks.
>
> **Not all SE-GNNs are restricted to TEs:** We highlighted multiple times that our focus is on SE-GNNs with the losses of Table 1 (Line 57 in Preliminaries, Statement of Thm 3.2, Limitations). Overall, the losses in Table 1 capture the notion of  “small, label-preserving explanations” in a relatively general fashion, offering insights into a wide SE-GNN family. Nonetheless, we appreciate the reviewers’ feedback, and we will clarify that our results pertain to a subset of all SE-GNNs also in the Abstract, Introduction, and Conclusions.
>
> **Rethink the term Trivial Explanation:** This paper aims to formalize SE-GNN explanation semantics, and the term 'Trivial Explanation' was meant to highlight simplicity, not negativity. On further reflection, we agree that “Trivial” may imply a bias. Therefore, we will rephrase it as 'Minimum Explanations' in the manuscript.
>
> **DC-GNN may overfit node features:** Note that SE-GNNs are considerably more expressive than our sparse side channel, therefore, it is much more likely that a standard SE-GNN will overfit to (potentially harmful) topological patterns rather than the side channel overfitting to node features [1]. Nonetheless, in DC-GNNs, both channels are jointly trained to maximize accuracy, and experiments show they balance well for generalization.
>
> [1] Graph Neural Networks Use Graphs When They Shouldn't
>
> **Add baselines beyond Table 1:** Following also the suggestion of rev. rjdA, we opted to include as a new baseline beyond Table 1, the model of Wu et al. 2020. Given the time and space constraints, we report the results for a subset of the datasets, leaving the full set of results to the revised manuscript.
>
> |BAColor|Accuracy|Channel|
> |-|-|-|
> |DIR| 98 +- 01 ||
> |DC-DIR| 98 +- 01 |Rule|
>
> |AIDS|Accuracy|Channel|
> |-|-|-|
> |DIR| 96 +- 03 ||
> |DC-DIR| 96 +- 03 |Rule|
>
> |BBBP|AUC|Channel|
> |-|-|-|
> |DIR| 64 +- 02 ||
> |DC-DIR| 65 +- 02 |Topo*|
>
> |MUTAG_0|Accuracy|Channel|
> |-|-|-|
> |see below|||
>
> Where “*” has the same meaning as is Table 3. Overall, these results are in line with those of other SE-GNNs, showing the generality of our proposal.
>
> **Add existing benchmarks:** We performed new experiments on MUTAG_0. The results are as follows:
>
> |MUTAG_0|Accuracy|Channel|
> |-|-|-|
> |GIN| 99 +- 1 ||
> |GSAT| 99 +- 1 ||
> |DC-GSAT| 99 +- 1 |Topo|
> |SMGNN| 99 +- 3 ||
> |DC-SMGNN| 98 +- 4 |Topo*|
> |GiSST| 99 +- 1 ||
> |DC-GiSST| 99 +- 1 |Topo|
> |DIR| 98 +- 2 ||
> |DC-DIR| 98 +- 2 |Topo|
>
> Model accuracy increases wrt the original MUTAG dataset, as expected (Serra & Niepert, 2022). Also, DC-GNNs prefer the topological channel as in MUTAG, and they match the performances of SE-GNNs. For BA-2MOTIFS, see **Q1**.
>
> **Show more critical distance to faithfulness:** Indeed, we show that explanations extracted by SE-GNNs can be misaligned wrt faithfulness metrics (Thm 5.2).  Whether those metrics are problematic in general is an interesting question beyond the scope of our work. We’ll add a sentence in Section 5 pointing to this issue.
>
> **Q1:** We opted for the Motif dataset in our experiments as, despite being designed with the same goal as BA-2MOTIFS, it is more challenging: it has three classes with three distinct motifs instead of two; it has more samples; and it comes with OOD splits.
>
> **Q2:** Our DC-GNNs are different from feature attribution methods in the form of the explanation they highlight. Feature attribution methods simply highlight the node features more relevant for prediction, whereas our DC-GNN can provide rules. In this sense, feature attribution methods cannot go “beyond” topological explanations, as the explanation would remain a subgraph with refined node features.
>
> Let us consider as an example RedBlueNodes, where graphs are of class 0 when they contain more red than blue nodes, encoded as one-hot vectors. There, a TE for an instance of class 0 contains a single red node, as it is the minimal subgraph allowing the SE-GNN to make the correct prediction. Then, running a feature attribution method can only highlight that the only relevant feature is the one related to the red color and that other features are irrelevant. However, this cannot give insight into the multiplicity of red vs blue nodes. Conversely, our DC-GNNs provide the rule *Red$\ge$Blue* as the explanation, which is more aligned with human expectations.
>
>
> **Q3:** Our theoretical analysis delineates tasks that are suitable and unsuitable for SE-GNNs and shows that simply aiming for alternative notions of explanations (like PI and faithfulness) can lead to large explanations and high computational complexity.
> Given these observations, a natural choice is to augment SE-GNNs with alternative modalities of explanations taking care of SE-GNNs’ limitations — and we believe DC-GNNs are a well-motivated step in this direction. We will clarify this link in the revised Abstract and Introduction.

---

> > ### Comment · Reviewer_3Jwx · 2025-04-02
> >
> > Thank you for your response. I appreciate the additional experimental results.
> >
> > Again, trivial, according to English dictionaries, means ignorable, of little significance or value, being the simplest possible case. The use of this word is misleading and has a negative connotation. Also, reading your abstract and introduction, a reader must believe that TEs are exactly those explanations extracted by SE-GNNs, which is incorrect.
> >
> > Conditional on the changes that a more suitable term will replace the term "trivial explanation" and that the abstract and intro are updated to make clear that TEs are those explanations extracted by *a subclass of* SE-GNNs, I will raise my score to 4.

---

### Official Review · Reviewer_Mh3n · 2025-03-14

**Overall Recommendation:** 4

**Summary:**

This paper discusses several definitions of Graph explanations and there internal connections. They provide analytical results on the connections between TEs, PIs, sufficient and necessary subgraphs explanations. Because they show that TEs are always less informative than other explanations & directly learning other explanations is not optimal, they construct DC-GNN that finds a middle ground of these two issues.

**Claims And Evidence:**

The paper draws connections between different subgraph explanations with rigorous proofs. I did not read the proofs fully but have examined the statement and proof sketch, which all look good to me. The second part of the paper focuses on the construction of DC-GNN and they have provided empirical results over 8 datasets.

**Essential References Not Discussed:**

N/A

**Experimental Designs Or Analyses:**

I have checked the setup and they are using widely-used datasets in other literature, and the evaluation metrics are related. A few questions / comments.

The paper mentions that GNNs, including DC-GNNs, can be computationally expensive. However, it does not provide a detailed analysis of the computational complexity or scalability of their approach. It would be beneficial to have a more thorough evaluation of the computational resources required by DC-GNNs, so we have a better sense if this can scale.

**Methods And Evaluation Criteria:**

This paper is well-motivated and I enjoy reading it much. Their analysis reveals that TEs align with PI explanations for a restricted but significant family of tasks.  However, in general, TEs can be less informative than PI explanations and are misaligned with accepted notions of faithfulness. Although synthetic datasets are easier to control (and we have groundtruth for subgraphs), it would be great to have more discussions and analysis on real-world datasets.

**Other Comments Or Suggestions:**

N/A

**Other Strengths And Weaknesses:**

One significant contribution is to analyze the information gain from using a simple TE subgraph, or more complicated PI and faithful subgraphs. The authors show that TE significantly contains less information despite of its simplicity and sometimes it is very unfaithful.

**Questions For Authors:**

Could you provide more details on the hyperparameter tuning process for DC-GNNs? How sensitive is the model's performance to different hyperparameter settings, and what steps have you taken to ensure the robustness of your results across a reasonable range of parameter values?

**Relation To Broader Scientific Literature:**

The analytical parts has bridged several different metrics in generating subgraph explanations. This paper provides a clear answer to what subgraph explanations are important and in what sense.

**Theoretical Claims:**

I have only checked proof sketch and they all look correct and sound to me.

---

> ### Author Rebuttal · Authors · 2025-03-31
>
> We thank the reviewer for their feedback. We are glad they enjoyed reading it and found it well motivated. Below, we address their remarks.
>
> **Computational resources for running DC-GNNs:** We would like to clarify a slight misunderstanding regarding our discussion of computational costs. Our argument mostly pertains to PIs being intractable to extract (Marques-Silva, 2023). In contrast, SE-GNNs aim to extract explanations with a single forward pass of the neural network, resulting in fast and tractable explanations. DC-GNNs, on their side, combine a standard SE-GNNs with a side channel implemented as a linear layer. The resulting complexity is primarily dominated by the SE-GNN-based channel, and the side channel adds negligible computational overhead. Below, we report an analysis of the run time for DC-GNNs compared to existing baselines, which will be integrated into the revised manuscript.
>
> |Graph-SST2|Time per epoch (seconds)|
> |-|-|
> |GSAT| 4.78 +- 0.23 |
> |DC-GSAT| 4.50 +- 0.32 |
> |SMGNN| 6.51 +- 0.37 |
> |DC-SMGNN| 6.32 +- 0.50 |
>
> |BBBP|Time per epoch (seconds)|
> |-|-|
> |GSAT| 0.55 +- 0.18 |
> |DC-GSAT| 0.58 +- 0.18 |
> |SMGNN| 0.53 +- 0.18 |
> |DC-SMGNN| 0.71 +- 0.22 |
>
> **More details about hyper-parameter tuning:**
>
> Hyper-parameters were optimized for the performance of the validation split. For datasets with OOD splits, only the ID validation split was used for tuning. We’ll clarify this in Appx. B.2.
>
> When choosing the values to test, we followed the standard practice of choosing values within a reasonable range, e.g., powers of $10$ (0.001, 0.01, 0.1, …), or other reasonable choices like $r \in \\{0.3, 0.5, 0.7, 0.9\\}$ for GSAT. When performance was satisfactory, we kept the default hyper-parameters of SE-GNN.
>
> We report below an additional analysis on the robustness to hyper-parameter selection. Given the time constraint in running those experiments, we focus only on representative hyper-parameters for DC-GSAT for the Motif dataset by aggregating values across 5 seeds. We plan to complement those results with the full experimental setting in the final revision of our manuscript.
>
>
>
>
> |(B)LEN hidden size|15|*30*|64|
> |-|-|-|-|
> |DC-GSAT| 92 +- 01 | 93 +- 01 | 93 +- 01 |
>
> |(B)LEN num layers|*2*|3|4|
> |-|-|-|-|
> |DC-GSAT| 93 +- 01 | 93 +- 01 | 93 +- 01 |
>
> |(B)LEN sigmoid temperature|0.1|*0.3*|0.5|
> |-|-|-|-|
> |DC-GSAT| 93 +- 01 | 93 +- 01 | 93 +- 01 |
>
> |GSAT’s $r$ parameter|0.5|*0.7*|0.9|
> |-|-|-|-|
> |DC-GSAT| 90 +- 02 | 93 +- 01 | 93 +- 01 |
>
>
> For comparison, we report in italic font the original value reported in the paper. The SE-GNN channel was the one selected by the training routine for all the experiments. Overall, results are stable across hyper-parameter choice, except for a small fluctuation when choosing $0.5$ as the parameter $r$ of GSAT. However, for $r \ge 0.7$, performance stabilizes.

---

> > ### Comment · Reviewer_Mh3n · 2025-04-06
> >
> > Thanks for the great responses here. Please update the paper with these results! My score remains.

---

### Official Review · Reviewer_oqKU · 2025-03-14

**Overall Recommendation:** 3

**Summary:**

This paper investigates Self-Explainable Graph Neural Networks (SE-GNNs) and their limitations in explanation quality. It formalizes the Trivial Explanations (TEs) generated by SE-GNNs and compares them with Prime Implicant (PI) and faithful explanations. The analysis shows that TEs align with PI explanations in some cases but are generally less informative and misaligned with faithfulness. To address this, the authors propose Dual-Channel GNNs, combining a white-box rule extractor with a standard SE-GNN. Experiments demonstrate that Dual-Channel GNNs can extract concise rules and perform better or on par with existing SE-GNNs.

**Claims And Evidence:**

The authors' claims are mainly threefold:
1. In Section 4, they formalize the trivial explanations (TEs) generated by SE-GNNs and compare them with Prime Implicant (PI) and faithful explanations.
2. In Section 5, they explore how Trivial Explanations can be unfaithful.
3. Based on the analysis and findings, in Section 6, they propose Dual-Channel GNNs, which combine a white-box rule extractor with a standard SE-GNN.

Both claims 1 and 2 are well-supported with theoretical evidence, while claim 3 is validated through extensive experiments demonstrating its effectiveness.

**Essential References Not Discussed:**

It seems that the relevant literature is sufficient.

Note: I am not familiar with this field.

**Experimental Designs Or Analyses:**

I believe that both qualitative and quantitative experiments can demonstrate the performance and interpretability of the proposed method in this paper.

**Methods And Evaluation Criteria:**

This paper focuses on Self-Explainable Graph Neural Networks (SE-GNNs) and aims to better understand the properties and limitations of their explanations.

First, in Section 4, the authors formalize the trivial explanations (TEs) generated by SE-GNNs and compare them with Prime Implicant (PI) and faithful explanations. Further, in Section 5, they explore how Trivial Explanations can be unfaithful. These discussions are accompanied by extensive theoretical evidence that addresses the problems described by the authors.

Based on their analysis and findings, in Section 6, the authors propose Dual-Channel GNNs, combining a white-box rule extractor with a standard SE-GNN. A large body of experiments demonstrates the effectiveness of the proposed method.

**Other Comments Or Suggestions:**

Please see "Other Strengths and Weaknesses".

**Other Strengths And Weaknesses:**

**Weaknesses or Suggestions:**

I believe the expression of this paper needs further improvement. Although the author's theory and techniques are sufficient, the paper lacks examples and a clear explanation of the domain background, making it difficult for readers unfamiliar with the self-explainable field or newcomers to the field to comprehend. This is particularly evident in the following points:

1. In the second paragraph of the introduction, the author does not clearly express that TEs (Trivial Explanations) are the key concept introduced in this paper. It is only later in the text that we realize this. This confusing logic may leave readers unsure of the relationship between TEs and existing PIs (Prime Implicants) and faithful explanations, leading them to believe that these are parallel categories of methods (perhaps with overlap). Furthermore, these concepts need to be illustrated with examples.

2. The author needs to explain the formulas in Table 1. While most readers are likely familiar with their meanings, there are unknown symbols in the table, and it is hard for us to understand what these symbols represent without further clarification.

3. The logic in Sections 2 and 3 needs improvement, especially in clarifying the motivation behind each concept and theory. For example, in Section 2, the introduction of logical classifiers is abrupt, lacking context and further explanation. Readers may struggle to understand its role and its connection to the paper, leading to confusion. Similar issues appear in Theorems 3.2, 3.4, and Remark 3.3. The author should note that this is not a technical report, and the content should not be simply listed but should follow a logical structure.

4. In the main text, each theory should be clearly explained with non-mathematical descriptions so that readers can understand its purpose and the motivation behind the author's derivations.

5. The author should provide an overview of the technical aspects of self-explainable GNNs, particularly in Sections 4, 5, and 6. This would help tie together the goals of each section. Currently, each section contains a lot of content, but without an overview, it is easy for readers to forget the purpose of each section and how they relate to the overall paper. It might seem like three unrelated technical modules are presented.

I will adjust my score based on the author's response.

**Questions For Authors:**

Please see "Other Strengths and Weaknesses".

**Relation To Broader Scientific Literature:**

I believe this paper could advance the interpretability of GNNs, but it is hard to say whether it will have a significant impact. In my view, the author's theory is solid, but the practicality of the proposed Dual-Channel GNNs will require time to be tested.

**Theoretical Claims:**

This paper is well-supported by theoretical evidence. I have checked all the proofs and did not find any explicit errors.

---

> ### Author Rebuttal · Authors · 2025-03-31
>
> Thank you for taking the time to review our work.  We are glad that you found it well supported by evidence and theoretically sound. We applied the following changes.
>
> **W1: Clarify TEs are a contribution** We revised the Introduction line 16 as follows:
>
> > Focusing on graph classification, we introduce the notion of Trivial Explanations (TEs) as the minimal subgraphs of the input that locally ensure the classifier outputs the target prediction. We then show that some popular SE-GNNs are implicitly optimized for generating TEs.
>
> **W1: TEs and PIs need examples** As an example,  Fig. 1 illustrates the different nuances between TEs, PIs, and faithfulness with a simple example of two different classifiers. We will make this connection clearer in the revised manuscript and bring this example to the forefront of the discussion in the Introduction. We also updated Fig 1’s description to ensure the figure is properly discussed and contextualized. This change could not be reported here due to space constraints.
>
> **W2:** We added a description of the formulas in Table 1. This change could not be reported here due to space constraints
>
> **W3: logical classifiers** In Section 2, different topics are introduced separately since they serve as background material for the main content. We updated the Logical classifiers paragraph as follows to make the context clearer:
>
> > To prove our theoretical results, we will use basic concepts from First Order Logic (FOL) as described in Barcelo and Grohe. [...] For ease of discussion, we fix two FOL classifiers that will be used to provide examples and intuition in the remainder of the paper: [...]
>
> **W3: Sections 3 lack context** We improved the contextualization of our results as follows:
>
> We added this in line 134 of Section 3:
>
> > Having established a link between SE-GNNs and TEs, we proceed to analyze the formal properties of TEs, starting from the following remark formalizing the notion of *informative explanation*.
>
> We revised line 142 of Section 3:
>
> > The following theorem, however, shows that TEs can fail to satisfy this desideratum for certain prediction tasks, indicating potential limits in the informativeness of TEs.
>
> We added this after Thm 3.4:
>
> > Intuitively, this result indicates that there exist two distinct graph classifiers for which TEs are the same for any input, meaning that by inspecting explanations alone, it is not possible to distinguish the two classifiers. Hence, TEs can fail to be informative wrt Remark 3.3
>
> **W4:** We list our modifications below, reporting only the cases where little intuition was provided and omitting the rest due to space constraints.
>
> After Thm 3.4:
>
> > This theorem shows cases in which, for two distinct classifiers, TEs match for any input graph, meaning that TEs are not informative for those classifiers wrt Remark 3.3.
>
> Line 132, column 2
>
> > Intuitively, Eq. 3 shows that the insight of Thm 3.4 applies even when aggregating TEs across all instances where the two classifiers yield the same prediction. Hence [...]
>
> After Thm 4.4
>
> > Thm 4.4 shows that PIs can overcome TEs' limits in certain tasks. For example, Thm 3.4's classifiers yield identical TEs but differing PIs (see Fig. 1)
>
> **W5: Technical Aspects of SE-GNNs** We added a deeper technical discussion in the Preliminaries and expanded Appx B.2 with more background on SE-GNN and a diagram akin to Fig. 1 in (Miao et al. 2022) showing a prototypical SE-GNN. Sections 4 and 5 remain independent of SE-GNN details, focusing on a theoretical analysis of TEs, PIs, and faithful explanations.
>
> **W5: Improve link between sections** We added the following sentences:
>
> Line 144 Section 4:
>
> > Having established the link between SE-GNNs and TEs in Section 3, in this section, we provide a comparative analysis between TEs and PIs for graph classifiers. [...]
>
> Line 193 in Section 4.1:
>
> > We now show that our previous observation that TEs equal PIs for $\exists x,y. E(x,y)$ generalizes all existential tasks. This reinforces the use of SE-GNNs in various practical applications and our proposed method (Section 6).
>
> Line 172 Section 4.2:
>
> > Although Section 4.1 shows TEs match PIs for existential tasks, real-world properties like counting cannot be expressed by existential formulas. Here, we show that beyond existential tasks, TEs can be less informative than PIs (Remark 3.3).
>
> Line 202 Section 4.2:
>
> > Thm 4.2  justifies SE-GNNs' performance on many benchmark datasets. However, Thm 3.4 and Thm 4.4 show that SEGNNs may not be informative for certain tasks, motivating an extension of SE-GNNs that preserves their performance on existential tasks but adaptively aids them in other tasks. We will exploit this observation in Section 6.
>
> Replace line 216 of Section 5 with:
>
> > This section reviews a general notion of faithfulness (Azzolin et al., 2024) in Definition 5.1 and shows that faithfulness, TEs (Section 3), and PIs (Section 4) can overlap – in some restricted cases – but are generally misaligned.

---

> > ### Comment · Reviewer_oqKU · 2025-04-05
> >
> > Thank you for your thoughtful responses and for clarifying the points I raised earlier. I now have a better understanding of the paper, which will positively impact my score.

---

### Decision · Program_Chairs · 2025-05-01

**Decision:**

Accept (poster)

**Comment:**

This paper offers a formal and insightful investigation into the explanatory capacity of Self-Explainable Graph Neural Networks (SE-GNNs), introducing the notion of Minimum Explanations and analyzing their relationship with Prime Implicant and faithful explanations. It further proposes Dual-Channel GNNs (DC-GNNs) as a practical solution to some of the theoretical limitations identified. The theoretical contributions are rigorous and well-motivated, and the empirical evaluation, while focused, provides evidence of the method’s effectiveness. The final scores reflect consensus toward acceptance, with three reviewers recommending acceptance and one leaning toward acceptance. I recommend acceptance.